# KinPFN: Bayesian Approximation of RNA Folding Kinetics using Prior-Data Fitted Networks

**Dominik Scheuer**[1][*]   **Frederic Runge**[1][*]   **Jörg K.H. Franke**[1]
**Michael T. Wolfinger**[2,3]   **Christoph Flamm**[2]   **Frank Hutter**[1,4]
[1]University of Freiburg, Germany   [2]University of Vienna, Austria
[3]RNA Forecast e.U., Vienna, Austria   [4]ELLIS Institute Tübingen, Germany
Correspondence to `runget@cs.uni-freiburg.de` or `dom.scheuer@gmail.com`.

## Abstract

RNA is a dynamic biomolecule crucial for cellular regulation, with its function largely determined by its folding into complex structures, while misfolding can lead to multifaceted biological sequelae. During the folding process, RNA traverses through a series of intermediate structural states, with each transition occurring at variable rates that collectively influence the time required to reach the functional form. Understanding these folding kinetics is vital for predicting RNA behavior and optimizing applications in synthetic biology and drug discovery. While *in silico* kinetic RNA folding simulators are often computationally intensive and time-consuming, accurate approximations of the folding times can already be very informative to assess the efficiency of the folding process. In this work, we present *KinPFN*, a novel approach that leverages prior-data fitted networks to directly model the posterior predictive distribution of RNA folding times. By training on synthetic data representing arbitrary prior folding times, *KinPFN* efficiently approximates the cumulative distribution function of RNA folding times in a single forward pass, given only a few initial folding time examples. Our method offers a modular extension to existing RNA kinetics algorithms, promising significant computational speed-ups orders of magnitude faster, while achieving comparable results. We showcase the effectiveness of *KinPFN* through extensive evaluations and real-world case studies, demonstrating its potential for RNA folding kinetics analysis, its practical relevance, and generalization to other biological data.

## 1 Introduction

Ribonucleic acid (RNA) plays a pivotal role in various biological processes, serving as a crucial intermediary between DNA and proteins while exerting significant regulatory functions through diverse mechanisms (Fu, 2014). Composed of four nucleotides – Adenine (A), Cytosine (C), Guanine (G), and Uracil (U) – the functionality of RNA is closely tied to its structure (Lodish et al., 2005): An RNA molecule adopts one or more native conformations that are essential for its biological activity (Fang et al., 2015). The dynamic process of how RNAs acquire their functional structure is known as the kinetic folding of RNA. During this process, the RNA strand transitions through several intermediate structural states, driven by intra-molecular interactions (Flamm et al., 2000; Yu et al., 2018). Since misfolding can lead to significant dysfunctions and diseases (Conlon & Manley, 2017), the study of RNA folding kinetics is highly relevant for biomedical applications.

An important aspect of folding dynamics is the study of the rates and pathways through which RNA molecules achieve their native structures (Chen, 2008). A common measure to quantify these processes are first passage times (FPTs), i.e. the time required to acquire a certain structure for the first time, and their cumulative distribution functions (CDFs) (Flamm et al., 2000; Wolfinger et al., 2004). These functions are derived from extensive simulations, requiring thousands of folding iterations to capture the probabilistic behavior of RNA molecules. While essential for understanding RNA dynamics, calculating FPT CDFs is computationally expensive (Wolfinger et al., 2004; Badelt et al., 2023), posing a significant barrier to real-time applications such as kinetic RNA design, which

---

[*]Equal Contribution.

is critical for drug discovery. While deep learning methods could improve the state of the art in RNA folding (Fu et al., 2022; Franke et al., 2024) and RNA design (Runge et al., 2024; Patil et al., 2024), they are not yet used in modeling RNA kinetics.

In this work, we present *KinPFN*, a novel deep learning-based approach that dramatically accelerates the computation of RNA first passage times. *KinPFN* leverages prior-data fitted networks (PFNs) (Müller et al., 2022) trained on synthetic datasets of RNA folding times to predict the entire CDF of folding times from just a few context examples in a single forward pass. By providing fast and accurate distribution approximations, *KinPFN* can be integrated with existing RNA kinetics simulators, offering comparable performance at a fraction of the computational cost. These speedups make KinPFN a valuable tool for the study of RNA folding kinetics, offering novel routes for applications in kinetic RNA design, which was previously intractable due to exponential runtimes of kinetic folding simulators, and promising fast analysis of RNA folding behaviors across multiple applications in drug discovery, medicine, biotechnology and synthetic biology.

Our main contributions are summarized as follows:

- We propose a new synthetic prior to sample datasets of RNA folding times. We use this synthetic data to train a prior-data fitted network to learn to predict the distribution of RNA first passage times, conditioned on a small set of context examples (Section 4.1).
- We introduce *KinPFN*, a new deep learning model for RNA kinetics. *KinPFN* provides accurate predictions of RNA first passage time distributions, accelerating kinetic simulations by orders of magnitude (Section 4.2).
- We evaluate *KinPFN*'s performance on synthetic and real-world RNA data (Section 5), demonstrating its practical utility through two case studies: an analysis of eukaryotic RNAs (Section 5.2) and a study of RNA folding efficiency (Section 5.3).
- In addition to its application to RNA folding kinetics, we assess *KinPFN*'s ability to generalize to different biological data sources by approximating gene expression data obtained from a previous smFISH (Femino et al., 1998; Raj et al., 2008) wet-lab analysis (Bagnall et al., 2020), demonstrating its potential to accelerate experimental protocols (Section 5.4).

We provide an overview of *KinPFN* in Figure 1. Our source code, data, and trained models are publicly available at `https://github.com/automl/KinPFN`.

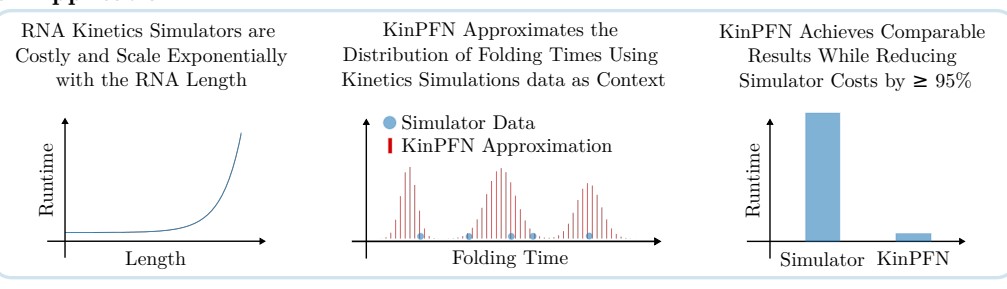

Figure 1: Graphical abstract. **a**: *KinPFN* is trained on synthetic RNA folding time distributions drawn from parameterized multi-modal Gaussians by minimizing the negative log-likelihood (NLL). **b**: *KinPFN* accelerates RNA kinetics simulators by predicting the RNA folding time distribution in a single forward pass, given a few folding times as context.

## 2 BACKGROUND

**RNA First Passage Times** Kinetic RNA folding is typically approximated by Monte-Carlo simulation techniques (Flamm & Hofacker, 2008). However, this is computationally expensive since enough stochastic simulations need to be accumulated to get a statistically representative time evolution of the state probabilities. Depending on the number of different structural states, which is typically huge, the path during folding, and the energy barriers between the states, the time to reach a certain structure for the first time, i.e., the *first passage time*, can differ across multiple kinetic simulations for a given RNA. By comparing the first passage time CDFs of different RNA molecules or under varying conditions, differences in the folding dynamics can be revealed and better understood. This comparison provides insights into the efficiency and stability of the different folding processes, examining the impact of various modifications, such as chemical alternations or evolutionary changes (Flamm et al., 2000). In this work, we use the term folding time as a synonym for first passage time and focus on RNA folding kinetics based on secondary structure information.

**Prior-Data Fitted Networks** Prior-data fitted networks (PFNs) (Müller et al., 2022) use a transformer-based model to perform approximate Bayesian inference. PFNs are trained to predict an output $y \in \mathbb{R}$, conditioned on an input $x$ and a training set $D_{\text{train}}$ of input-output pairs. During training, these samples are drawn from a prior distribution over datasets $p(\mathcal{D})$, optimizing the Cross-Entropy loss for a PFN $q_\theta$ with parameters $\theta$,

$$\ell_\theta = \mathbb{E}_{(x,y) \cup D_{\text{train}} \sim p(\mathcal{D})} \left[ -\log q_\theta(y \mid x, D_{\text{train}}) \right], \tag{1}$$

for predicting the label $y$, given $x$ and $D_{\text{train}}$. As shown by Müller et al. (2022), this approach directly minimizes the Kullback-Leibler (KL) divergence between the prediction of the PFN and the true posterior predictive distribution when training on many samples of the form $(x, y) \cup D_{\text{train}}$. In this work, we adapt this strategy to tackle the prediction of RNA first passage time distributions, accounting for the specific challenges of the probabilistic behavior of RNA molecules that is also reflected in kinetic simulators by renouncing quantile information. For more details, see Section 4.2.

## 3 RELATED WORK

*In silico* analysis of RNA folding kinetics can be divided into nucleotide-resolution and coarse-grained approaches. While the first yields a high level of simulation details, the latter typically allows studying larger systems, i.e. longer RNA chain lengths. The first publicly available tool for computing RNA folding kinetics at nucleotide resolution is *Kinfold* (Flamm et al., 2000), a Markov-chain Monte Carlo (MCMC) method that is still considered one of the most accurate approaches available (Fukunaga & Hamada, 2019). This accuracy, however, comes at the cost of runtime as *Kinfold* MCMC simulations typically require a large number of trajectories to obtain reliable results. While it is possible to simulate the folding kinetics of RNA chains of several hundreds of nucleotides, such calculations require substantial compute (Fukunaga & Hamada, 2019). This limitation inspired accelerating techniques like memoization and parallelization (Aviram et al., 2012), or shortcuts for the energy calculations of RNA secondary structures as implemented in *Kfold* (Dykeman, 2015). In contrast, we develop *KinPFN* as an extension to existing kinetic RNA folding simulators to massively speed up every kinetic simulator that produces first passage times.

An Alternative to *KinPFN* are probabilistic density estimators like kernel density estimation (KDE) (Bishop, 2006), Gaussian Mixture Models (GMM) (Bishop, 2006) or Bayesian Gaussian Mixture Models, also known as Dirichlet Process GMMs (DP-GMM), which utilize a Variational Bayesian estimation of Gaussian mixtures (Blei & Jordan, 2006). Similar to *KinPFN*, GMM and DP-GMM aim to model the posterior predictive distribution as a multi-modal Gaussian distribution. While GMMs struggle with complex data structures, especially when the number of modes is unknown, Bayesian approaches like DP-GMM can dynamically adjust the number of mixture components (McLachlan et al., 2019; Neal, 2000). Alternatively, kernel density estimation (KDE) offers a non-parametric approach by estimating probability densities through the summation of kernels, like Gaussians, over data points (Bishop, 2006). From a deep learning perspective, methods based on normalizing flows (Rezende & Mohamed, 2015), variational autoencoders (VAEs) (Kingma, 2013), or a probabilistic transformer as proposed in Franke et al. (2022), would be well suited for probability density estimation of RNA folding kinetics. However, these methods typically require large amounts of training data which is not available for RNA folding kinetics. Instead, we approach the

problem of folding time prediction using a synthetic prior to train a PFN for direct approximation of the CDF of folding time distributions.

While, to the best of our knowledge, *KinPFN* is the first deep learning approach for RNA folding kinetics, PFNs were previously applied to multiple problems like few shot image classification (Müller et al., 2022), classification for small tabular datasets (Müller et al., 2022; Hollmann et al., 2023), extrapolation of learning curves (Adriaensen et al., 2023), Bayesian optimization and hyperparameter optimization (Müller et al., 2023; Rakotoarison et al., 2024), and time series forecasting (Dooley et al., 2024). For more discussions on related work, please see Appendix A.

## 4    APPROXIMATION OF RNA FOLDING TIME DISTRIBUTIONS

We consider the problem of learning the posterior predictive distribution (PPD) of first passage times for an RNA molecule $\phi \in \{A, G, C, U\}^l$ of length $l$, conditioned on a small set of initial examples, to approximate the cumulative distribution function (CDF). Formally, the first passage time $t$ is the time required for the RNA $\phi$ to fold from an initial structure $\omega_{\text{start}}$ into a stop structure $\omega_{\text{stop}}$ while transitioning through arbitrary intermediate structural states. Running $M$ folding simulations under the same conditions (for RNA sequence $\phi$, $\omega_{\text{start}}$, and $\omega_{\text{stop}}$) yields distinct first passage times $t_1, \ldots, t_M$. By aggregating these times, we compute the fraction of molecules $\phi$ folded by time $T$, denoted $F^\phi(T)$, where $F_t^\phi(T) = P(t \leq T)$ represents the CDF of the stochastic variable $t$.

The problem we consider in this work can be formulated as follows: Given $N \ll M$ observed first passage times $t_1, \ldots, t_N$ and a prior distribution over first passage times from which we can generate samples, we aim to approximate the PPD $q(t \mid t_1, \ldots, t_N)$. With an approximated PPD, we can compute the predicted CDF $\hat{F}^\phi(T)$, which approximates the true CDF $F^\phi(T)$; the fraction of molecules folded by time $T$.

In the following sections, we describe our approach to define a synthetic prior of first passage time distributions that allows us to approximate the PPD of folding times (Section 4.1) and explain the development of *KinPFN* in detail (Section 4.2).

### 4.1    A SYNTHETIC PRIOR FOR RNA FOLDING TIME DISTRIBUTIONS

Obtaining large amounts of prior RNA kinetics data to train a deep learning model, particularly for longer RNAs, is currently infeasible due to the exponential runtime of accurate kinetic simulators (see Figure 7 in Appendix B). This hinders us from using traditional Bayesian approaches for the approximation of RNA first passage times, e.g., by training a variational autoencoder (VAE) (Kingma, 2013). Therefore, we take an alternative approach, training a PFN solely on a synthetic prior of RNA first passage time distributions. However, developing a synthetic prior for molecular problems is challenging since it seems impossible to generate meaningful synthetic combinations of molecule features with posterior information from a process depending on these features. We, therefore, develop *KinPFN* independent of molecular features and restrict its input to first passage times only. This offers the advantage that we can apply *KinPFN* to predict first passage time distributions at test time, independent of the underlying data-generating process.

For the development of our synthetic FPT prior, we leverage the observation that RNA first passage time distributions often exhibit CDFs with regions of slower growth interspersed with steeper transitions, leading to distinct plateaus and multiple changes between convex and concave sections representing inefficiencies in the corresponding folding pathway (Flamm et al., 2000; Wolfinger et al., 2004). These patterns make multi-modal distributions a natural choice to model the complexity of such processes synthetically, as they are designed to capture data with multiple local maxima or modes (Hartigan & Hartigan, 1985). We thus construct a prior distribution over RNA first passage times as a family of multi-modal Gaussian distributions $\{P_{\psi_k} \mid k \in \{2, 3, 4, 5\}, \psi_k \in \Psi_k\}$. Each multi-modal distribution in this family comprises $k$ Gaussian components, each characterized by its own mean $\mu_i$ and standard deviation $\sigma_i$, $i = 1, \ldots, k$. The parameter space $\Psi_k$ thus defines the family of distributions, with each specific distribution parameterized by a vector $\psi_{\mathbf{k}} = ((\mu_1, \sigma_1), (\mu_2, \sigma_2), \ldots, (\mu_k, \sigma_k))$ within $\Psi_k$. We illustrate a synthetic bi-modal PDF alongside its corresponding CDF and examples of synthetic first passage time CDFs in Figure 8.

Since we cannot make any further assumptions about the distribution of folding times, especially when generating synthetic data, $x$ and $y$ of a prior distribution $p(\psi_k)$ are considered completely independent. Consequently, we decide to assign a value of zero to all variables $x$, representing no prior information, while the $y$ variables are ultimately sampled from the aforementioned multi-modal distributions. As the targets $y$ represent synthetic first passage times, they will be referred to as $t$ from this point forward. We set the range of possible first passage time values $t \sim p(\psi_k)$ to $[10^{-6}, 10^{15}]$, a range that covers a large fraction of possible folding processes based on observations from preliminary kinetic simulations. To mimic realistic first passage time distributions, we choose bounded uniform base means $\mu_i^{\text{base}} \sim \mathcal{U}(-5, 16)$, and uniformly distributed standard deviations $\sigma_i \sim \mathcal{U}(0.1, 4.2)$ based on preliminary experiments. To increase the variability of the prior, we introduce a uniformly distributed shifting parameter $\delta \sim \mathcal{U}(-6, 15)$, which is sampled only once and fixed for all $i = 1, \ldots, k$. The final means $\mu_i$ are then given by:

$$\mu_i = \mu_i^{\text{base}} + \delta \qquad , \tag{2}$$

with the probability density function (PDF) of the multi-modal Gaussian distribution parameterized by $\psi_k$ expressed as

$$p(\psi_k, x) = \sum_{i=1}^{k} \exp\left(-\frac{(\log x - \mu_i)^2}{2\sigma_i^2}\right) \qquad , \tag{3}$$

for a value $x$.

To sample first passage times (FPTs) from these PDFs, we generate the PDF over a logarithmically spaced range of $x$-values within the provided FPT bounds and employ the inverse transformation method, known as the Smirnov transformation. The required series of calculations to derive the CDF, its quantile function $\text{CDF}^{-1}$, different normalizations to properly scale the functions, and logarithmic transformations are detailed in Appendix C.1. The prior distribution over synthetic RNA first passage times used in this work is then represented by the log-encoded samples from a multi-modal Gaussian distribution $p(\psi_k) \in P_{\psi_k}$:

$$\mathbf{Y} = \log_{10}\left(\left\{\text{CDF}^{-1}(\psi_k)\left(\mathcal{U}(0, 1)\right) \mid p(\psi_k)\right\}\right). \tag{4}$$

## 4.2 PFNs for the Approximation of RNA Folding Time Distributions

We propose to use PFNs (Müller et al., 2022) to accelerate kinetic simulations for RNA first passage time distributions. During training, the PFN $q_\theta$ with model parameters $\theta$ is presented with $M$ synthetic first passage times, $\{(0_i, t_i)\}_{i=1}^{M}$, sampled from the prior distribution $p(\psi_k)$. To enable the model to generalize across varying amounts of training data instead of a fixed number of context folding times, this example set is split at a random cutoff point $N \sim \mathcal{U}(0, M - 1)$, resulting in a training subset $D_{\text{train}} = \{(0_i, t_i)\}_{i=1}^{N}$, while the remaining first passage times are held out via masking. These held-out times, $t_{\text{test}} = \{t_{N+1}, \ldots, t_M\}$, are then used as targets for prediction by minimizing the prior-data negative log-likelihood (NLL) according to Equation 1:

$$\ell_\theta = \mathbb{E}_{(0, t_{\text{test}}) \cup D_{\text{train}} \sim p(\psi_k)} \left[-\log q_\theta(t_{\text{test}} \mid 0_{\text{test}}, D_{\text{train}})\right] \qquad . \tag{5}$$

Figure 2 schematically illustrates this training process of *KinPFN* for a single batch of size $B$, along with its application in approximating the posterior predictive distribution (PPD) of RNA first passage times using $N$ real folding times as context obtained from a kinetic simulator.

***KinPFN* Architecture and Hyperparameters** We adopt the transformer-based (Vaswani et al., 2017) PFN architecture as proposed by Müller et al. (2022) and treat each pair $(0, t)$ as a separate token. To learn the distribution of the targets rather than their specific ordering, we deliberately omit positional encoding to maintain permutation invariance according to Müller et al. (2022). Since the first passage times $t$ have already been log-encoded to the range $[-6, 15]$ in the prior distribution $p(\psi_k)$ (see Section 4.1), we encode the input with a linear layer after normalizing the data to zero mean and a standard deviation of one while preserving the distributional properties. Following Müller et al. (2022), we mask the attention matrix s.t. each position only attends to the training positions. This ensures that only training examples influence each other while test samples remain independent. We use the Adam optimizer (Kingma & Ba, 2015) with a cosine decay (Loshchilov & Hutter, 2017) and a linear learning rate warm-up over 25% of the training steps as previously

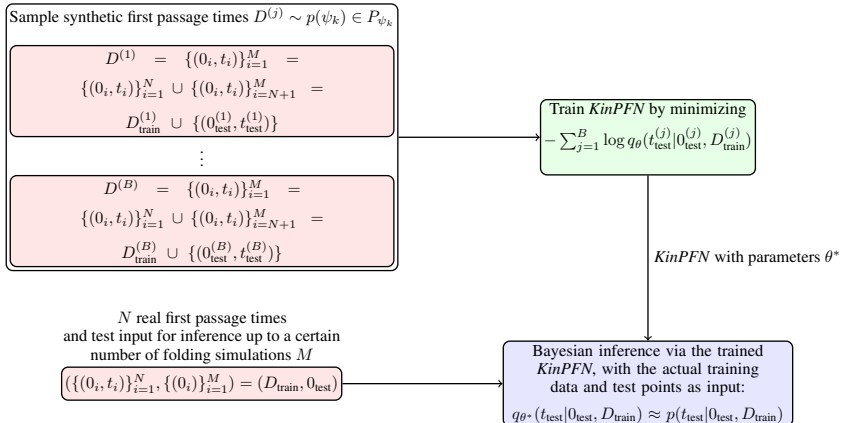

Figure 2: A schematic visualization of *KinPFN*. Diagram based on Müller et al. (2022).

proposed (Müller et al., 2022; Adriaensen et al., 2023). *KinPFN* outputs a discretized distribution $q_\theta(t|0, D_{\text{train}})$ (*Riemann distribution*; see Müller et al. (2022)) using a finite number of buckets with equal likelihood of containing $t$; a hyperparameter that is included in our hyperparameter optimization (HPO) procedure leading to a final number of 1,000 buckets for *KinPFN*, initialized on a batch of 100,000 prior samples. A visualization of the discretized distribution $q_\theta$ can be found in Appendix H.4. Further hyperparameters, like the number of layers, the embedding size, or the learning rate are inherited from the Transformer architecture. Given the infinite nature of synthetic training data, we set the dropout rate and the weight decay to zero. We tune hyperparameters in two separate runs using Neural Pipeline search (NePS) (Stoll et al., 2023). More details regarding hyperparameters, hyperparameter optimization, and the final configuration of *KinPFN* can be found in Appendix D. The final model of *KinPFN* was trained for roughly five hours on a single A40 GPU.

## 5 Experiments

*KinPFN* was trained on synthetic datasets of RNA folding times to learn to predict the distribution of first passage times, conditioned on a few examples. Therefore, the predictions only depend on example folding times for a given RNA but not on other features, e.g., its length, sequence composition, structure, or energy parameters. In this section, we show that this feature of *KinPFN* is a main contributor to its practical relevance. First, we confirm its ability to transfer from the synthetic prior data to realistic scenarios using a test set of simulations for randomly generated RNAs (Section 5.1). Then, we demonstrate the practical importance of *KinPFN* in two case studies: We show that *KinPFN* is capable of approximating first passage time distributions of natural RNAs (Section 5.2) and analyze the folding efficiency of different RNA sequences (Section 5.3). Finally, we assess *KinPFN*'s ability to generalize to different biological data by approximating gene expression data from a previous study (Bagnall et al., 2020) (Section 5.4). Preliminary evaluations for the predictions on samples from the synthetic prior are shown in Appendix H.1. We report performance in terms of prior-data negative log-likelihood (NLL) between the approximated posterior predictive distribution (PPD) and the true first passage time distribution and mean absolute error (MAE) between the CDF of the approximated PPD $\hat{F}(t)$ and the true target CDF $F(t)$. More information about these measures can be found in Appendix F. All experiments analyzing runtimes were benchmarked on a single AMD Milan EPYC 7513 CPU with 2,6 GHz.

### 5.1 *KinPFN* Transfers to Real-World Scenarios

We assess the general capabilities of *KinPFN* to transfer from synthetic data to data obtained from kinetic simulators. In particular, we analyze the robustness of *KinPFN* to changes in the sequence length of the RNA, the start and stop structure, and different kinetic simulators. To do so, we create a novel test set of 635 randomly generated RNA sequences with lengths between 15 and 147 nucleotides, run *Kinfold* (Flamm et al., 2000) for 1,000 simulations on each of the test samples and extract first passage times (FPTs) from the simulations. We compare *KinPFN* to GMMs and DP-

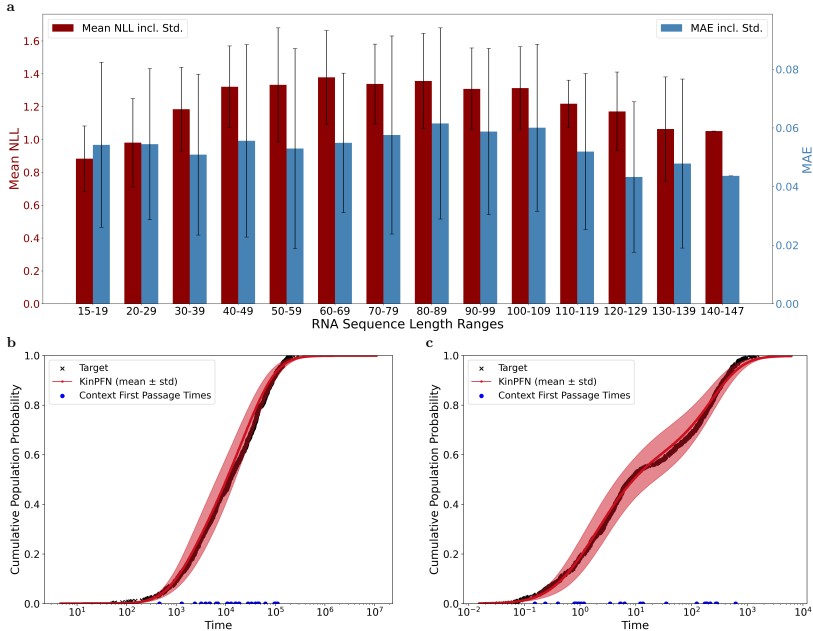

Figure 3: *KinPFN* approximations of first passage time distributions for simulation data of random RNA sequences across different settings. **a**: *KinPFN* testing set PPD mean NLL losses along with the CDF MAEs across RNA sequence length ranges. Error bars show the standard deviation of the losses. **b**: Example approximation for an alternative folding path of a 75 nucleotide RNA sequence with ground truth data obtained from *Kinfold* simulations. **c**: Example approximation for a 56 nucleotide RNA using *Kfold* simulation data as ground truth. We use $N = 25$ context first passage times for all experiments. Approximation examples show the mean and standard deviation around the mean for 20 predictions with different context examples sampled at random.

GMMs across multiple modes as well as KDE on this dataset, evaluating their performance across varying amounts of context using identical context first passage times. The modes we use for the GMMs align with our assumption in the synthetic prior for *KinPFN* (Section 4.1). Information about hyperparameter optimization for the competitors can be found in Appendix E. To analyze the performance of *KinPFN* for arbitrary folding paths that do not include the unfolded or minimum free energy structural states, we additionally run *Kinfold* on a randomly generated RNA sequence of 75 nucleotides and predict the PPD of first passage times for alternative folding paths. For the evaluation of *KinPFN*'s robustness to changes of the simulator, we use the *Kfold* (Dykeman, 2015) kinetic simulator to obtain FPTs for a randomly generated RNA of length 56. We provide more details about our novel test set in Appendix G. Predictions with different context lengths, more competitor evaluations, and results for additional RNAs are reported in Appendix H.2 and H.3.

**Results** Table 1 provides a comparison of *KinPFN* with the GMM, DP-GMM and KDE on our introduced test set with respect to NLL. Please find results for mean absolute error (MAE) and Kolmogorov-Smirnov (KS) statistic as well as explanations about these metrics in Appendix H.2 and F, respectively. Across all three metrics, *KinPFN* consistently demonstrates lower mean losses from a sample size of 25 onwards, outperforming the other approaches across various context first passage times ($N \in \{25, 50, 75, 100\}$). Consistent with our expectations, the performance of *KinPFN* constantly improves with more context FPTs. For a context size of ten, *KinPFN* performs slightly worse than KDE in terms of MAE and KS, achieving the second best performance while still outperforming KDE in terms of NLL. However, while we do observe visually strong approximations with a context size of ten in later experiments with *KinPFN* (see e.g. Section 5.3), we note that these results should be taken with care due to relatively large KS values, indicating that the predicted distributions do not strongly match with the ground truth distributions at a context size of ten. As shown in Figure 3a, *KinPFN* performs well across all sequences of the test set independent of the sequence length, given only 25 context points.

This is an important finding since especially simulations for long RNAs could benefit from accelerations with *KinPFN*. Similarly, we observe a very good fit of the approximation of the CDF of first passage times for folding paths between alternative structures (Figure 3b) and the application of *KinPFN* to simulations obtained from *Kfold* instead of *Kinfold* (Figure 3c). Our results thus indicate that *KinPFN* seems to generalize across different sequence lengths, start and stop structures, and different simulators. Notably, the approximations with *KinPFN* only require 2,5% of the compute budget of the original simulators to achieve comparable results. However,

Table 1: Evaluation of *KinPFN*, *KDE*, and multiple *GMM$_k$* and *DP-GMM$_k$* models with different initial modality assumptions $k \in \{2, 3, 4, 5\}$ on a newly introduced testing set comprising 635 real-world first passage time distributions in terms of prior-data negative log-likelihood loss (lower is better) with context first passage time cutoffs $N \in \{10, 25, 50, 75, 100\}$.

| Method | First Passage Times $N$ | | | | |
|---|---|---|---|---|---|
| | 10 | 25 | 50 | 75 | 100 |
| *KinPFN* | **1.3739** | **1.2435** | **1.2047** | **1.1916** | **1.1858** |
| *GMM$_2$* | 2.3122 | 1.3612 | 1.2355 | 1.2036 | 1.1933 |
| *GMM$_3$* | 5.2469 | 1.5830 | 1.2838 | 1.2132 | 1.1910 |
| *GMM$_4$* | 13.1325 | 1.9922 | 1.3676 | 1.2480 | 1.2119 |
| *GMM$_5$* | 37.5845 | 2.7708 | 1.4957 | 1.2953 | 1.2374 |
| *DP-GMM$_2$* | 1.6285 | 1.3529 | 1.2618 | 1.2305 | 1.2150 |
| *DP-GMM$_3$* | 1.6268 | 1.3549 | 1.2653 | 1.2323 | 1.2155 |
| *DP-GMM$_4$* | 1.6294 | 1.3558 | 1.2663 | 1.2337 | 1.2169 |
| *DP-GMM$_5$* | 1.6256 | 1.3572 | 1.2675 | 1.2337 | 1.2175 |
| *KDE* | 1.4370 | 1.2559 | 1.2133 | 1.2003 | 1.1957 |

the accuracy of the *KinPFN* approximations across all experiments can be further improved as we observe better performance with an increasing number of context examples (see also Table 8, 9, and 10 in Appendix H.2). This, however, comes at the cost of additional simulator runtime.

## 5.2 *KinPFN* APPROXIMATES FIRST PASSAGE TIMES OF EUKARYOTIC RNAS

While we observed robust performance of *KinPFN* for randomly generated RNA sequences, predictions for natural RNAs might be more challenging. In particular, highly structured RNAs like transfer RNAs (tRNA) or ribosomal RNAs (rRNA) might show different folding behavior compared to random RNA sequences due to million years of evolutionary pressure (Vicens & Kieft, 2022; Herschlag, 1995). We, therefore, decide to evaluate *KinPFN* on a tRNA[phe] of 76 nucleotides (RNAcentral Id: URS000011107D_4932) and a 5S rRNA of 121 nucleotides (RNAcentral Id: URS000055688D_559292) from *Saccharomyces cerevisiae*, one of the most extensively studied eukaryotic model organisms in molecular and cell biology, commonly known as brewer's yeast. For our experiments, we again use 1,000 *Kinfold* simulations as the ground truth data.

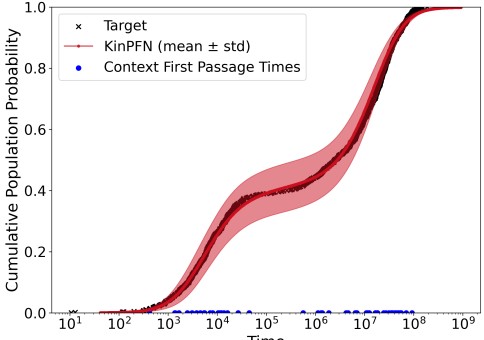 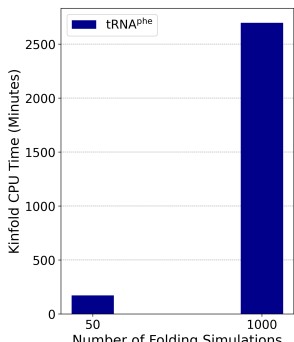

Figure 4: *KinPFN* first passage time CDF approximations for *Saccharomyces cerevisiae* tRNA[phe]. We show the mean and standard deviation for 20 predictions of *KinPFN*, each using 50 randomly sampled context times (left). On the right side, we show the runtime of *Kinfold* for 50 and 1,000 kinetic simulations for the tRNA[phe].

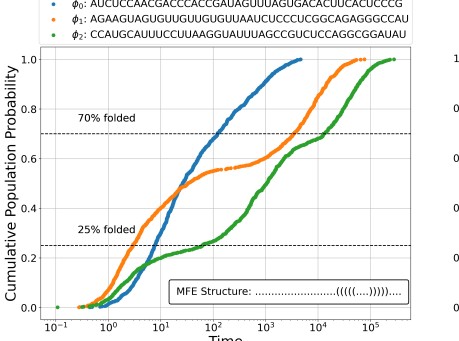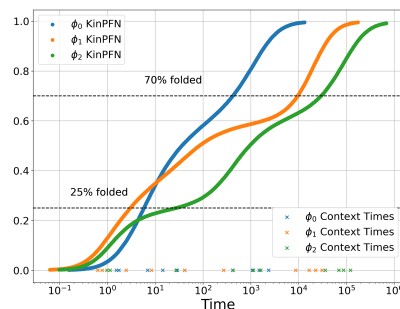

Figure 5: RNA folding efficiency analysis. The left plot shows the ground truth CDFs $F(t)$ for three sequences $\phi_0$, $\phi_1$ and $\phi_2$, representing the fraction of molecules folded into the MFE conformation (shown in dot-bracket notation (Hofacker et al., 1994)) over time $t$. The right plot displays the *KinPFN* approximations $\hat{F}(t)$ with ten *Kinfold* times as context.

**Results**  Figure 4 shows the first passage time CDF approximations of *KinPFN* for the tRNA[phe] (left). We observe that *KinPFN* is capable of approximating the ground truth data nearly perfectly using only 50 context first passage times. The runtime plot in Figure 4 (right) visualizes the decrease of the computational demands as the approximations of *KinPFN* result from using only 5% of the original compute budget, reducing the required CPU time from approximately 2,686 minutes (1000 simulations) to 170 minutes (50 simulations) while achieving nearly identical results. More predictions with different context times, as well as similar results for the 5S rRNA and further RNA types, are shown in Appendix H.5 and H.6. We conclude that *KinPFN* is capable of accurately approximating the CDFs of first passage times for real-world, structured RNAs like tRNA and rRNA.

## 5.3 CASE STUDY: RNA FOLDING EFFICIENCY ANALYSIS

To demonstrate the utility of *KinPFN*, we conduct a case study focused on comparing the folding efficiency of three 43 nucleotide long RNA molecules ($\phi_0$, $\phi_1$, $\phi_2$) that are predicted to fold into the same minimum free energy (MFE) structure. Alterations in the RNA sequences, such as mutations or modifications – often driven by evolutionary optimization – can have a significant effect on the folding dynamics (Flamm et al., 2000). A comparison of the CDFs of first passage times can distinguish molecules that fold more or less efficiently and provide information about how alternations in the molecules impact the folding behavior, an important aspect for RNA-based therapeutics (Mollica et al., 2022). For our experiment, we simulate 1,000 folding trajectories from the open chain to the MFE structure using *Kinfold* and calculate the ground truth first passage time CDFs shown in the left plot of Figure 5 for each of the 3 RNA molecules.

**Results**  We find that *KinPFN* captures the general folding behavior of the RNAs accurately, as shown in Figure 5 (right). However, while it captures the saddle points of the CDFs of $\phi_1$ (orange) and $\phi_2$ (green) arguably well, it is slightly less accurate for the most efficiently folding RNA, $\phi_0$ (blue). Remarkably, the *KinPFN* approximations were obtained using only ten context times, marking a 100× speed-up compared to each of the three individual simulation trajectories. Results for more approximations using different context lengths are shown in Appendix H.7.

## 5.4 *KinPFN* GENERALIZES TO GENE EXPRESSION DATA

Besides their usage in RNA folding kinetics analysis, CDFs of different distributions are a common tool for the analysis of biological data. For example, Bagnall et al. (2020) analyzed the messenger RNA (mRNA) expression of interleukin-1-$\alpha$ (*IL-1$\alpha$*), interleukin-1-$\beta$ (*IL-1$\beta$*), and tumor necrosis factor-alpha (*TNF-$\alpha$*) to study inducible gene expression in the immune toll-like receptor (TLR) system. Using single-molecule fluorescence *in situ* hybridization (smFISH) (Femino et al., 1998; Raj et al., 2008) analysis of the cumulative probability distribution of *IL-1$\alpha$*, *IL-1$\beta$*, and *TNF-$\alpha$* mRNA expression in two cell lines (established RAW 264.7 macrophage cells and bone-marrow-derived macrophages (BMDM)) stimulated with lipid A, Bagnall et al. (2020) demonstrate conserved vari-

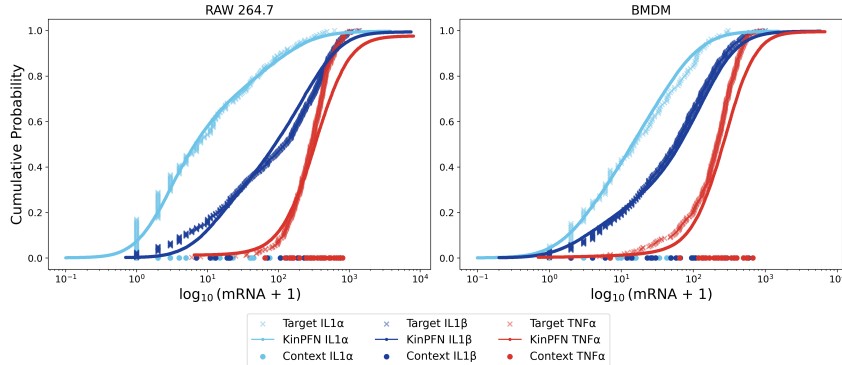

Figure 6: Approximation of mRNA expression of *IL-1α*, *IL-1β* and *TNF-α* in RAW 264.7 and BMDM cells. We plot approximations using only 25 context data points per gene.

ability in the TLR system across cell types, suggesting different modes of regulation of *IL-1β* and *TNF-α* expression. We use this experiment to analyze the capability of *KinPFN* to generalize to different biological data. Specifically, we use the raw count data of 447, 718, and 356 RAW 264.7 and 447, 732, and 322 BMDM cells for *IL-1α*, *IL-1β*, and *TNF-α*, respectively, to predict the cumulative probability functions of mRNA expression to replicate the outcome of the smFISH experiment of Bagnall et al. (2020) with *KinPFN* while using only a fraction of the data.

**Results** Figure 6 illustrates the approximations of the mRNA expression of *IL-1α*, *IL-1β*, and *TNF-α*. We observe that *KinPFN* can approximate the gene expression with high accuracy, using only roughly 8% of the expression data. These results suggest that – besides its application to RNA folding kinetics – *KinPFN* could be a valuable tool for different types of analysis across biological questions, including the potential to speed up even wet-lab experiments (see also Appendix H.8).

## 6 CONCLUSION, LIMITATIONS & FUTURE WORK

We present *KinPFN*, the first work that uses prior-data fitted networks for biological data. Trained on a synthetic prior, we show that our novel approach can accurately model RNA folding kinetics while accelerating RNA first passage time analysis by orders of magnitude. Moreover, we demonstrate that *KinPFN* generalizes to gene expression data obtained from wet-lab smFISH analysis, suggesting that *KinPFN* could be applicable to the analysis of a wide range of different biological questions.

**Limitations** While showing impressive accuracy across multiple tasks, *KinPFN* also has limitations. Since it is purely trained on synthetic first passage time data, it depends on a data-generating approach like kinetic simulators during inference. Consequently, *KinPFN*'s performance is bounded by the accuracy of the simulator. Incorporating other features, like the RNA sequence, structure, or energy information, could mitigate this issue. However, it is an open problem to implement the required information in a synthetic prior without using external data sources. Additionally, *KinPFN* would benefit from larger-scale evaluations, e.g., on longer RNAs, to confirm its independence of RNA features like sequence length. However, obtaining this kind of data is currently infeasible due to the large computing demands of available simulators and the problem's complexity. Further, *KinPFN* is limited to a bounded range of time values; however, so far, we have not experienced this limitation to be a major problem, and the training time of *KinPFN* is moderate, allowing retraining on adapted ranges. Similar to GMMs and KDEs, the performance of *KinPFN* strongly depends on the provided context. We tried to compensate for that by showing mean and standard deviation around the mean across 20 context inputs to quantify the variation in *KinPFN* approximations.

**Future Work** Using synthetic data for biological applications appears very promising. Unlike GMMs or standard KDEs, *KinPFN* is not limited to predefined kernels or Gaussian distributions; we consider the definition of synthetic priors using different distributions as future work. Generally, PFNs could play an important role in the field of structural biology, with the potential to substantially impact biological analysis, offering tremendous possibilities to accelerate scientific discovery.

REPRODUCIBILITY STATEMENT

To ensure the reproducibility of our results, we have made our source code, the trained model, and datasets publicly available at `https://github.com/automl/KinPFN`. The repository contains detailed instructions for setting up the required conda environment and package installs (see README.md). Model checkpoints of *KinPFN* are provided in the `models` directory. The validation and test sets are stored in the `neps_validation_set` and `kinpfn_testing_set` directories, respectively. We provide notebooks (along with the required experiment data) to demonstrate the training and evaluation of *KinPFN* and for reproducing results in the `notebooks` directory. We recommend using a single GPU with at least 48GB of memory for training *KinPFN*. However, for inference, a single CPU should be sufficient. Following the provided instructions, it should be straightforward to reproduce our environment, train and evaluate *KinPFN*, and replicate our experiments with minimal effort.

AUTHOR CONTRIBUTIONS

F.R. conceptualized the study and developed the methodology. D.S. and F.R. wrote the manuscript, designed figures, and were responsible for data curation. D.S. implemented the model and conducted all experiments. J.F. contributed to the experimental design and baseline selection. M.W. and C.F. provided expertise in RNA kinetics simulations and theory. J.F., M.W., and C.F. assisted with manuscript refinement and figure layout. F.H. provided project supervision and secured funding.

ACKNOWLEDGMENTS

Dominik Scheuer and Frederic Runge would like to thank Samuel Müller and Steven Adriaensen for helpful discussions and valuable comments. This work is supported in part by the European Union's Horizon Europe Doctoral Network programme under the Marie-Skłodowska-Curie grant agreement No 101072930 (TACsy), the Novo Nordisk Foundation grant NNF21OC0066551 (MATOMIC), and the Austrian Science Fund FWF grant I-6440 N. The authors further acknowledge funding by the German Research Foundation (DFG) under SFB 1597 (SmallData), grant no. 499552394, and through grant no. 417962828 as well as support by the state of Baden-Württemberg through bwHPC and the German Research Foundation (DFG) through grant no INST 39/963-1 FUGG (bwForCluster NEMO) and grant INST 35/1597-1 FUGG (bwForCluster Helix). Frank Hutter acknowledges the financial support of the Hector Foundation. This research was funded by the European Union (via ERC Consolidator Grant DeepLearning 2.0, grant no. 101045765). Views and opinions expressed are however those of the author(s) only and do not necessarily reflect those of the European Union or the European Research Council. Neither the European Union nor the granting authority can be held responsible for them.

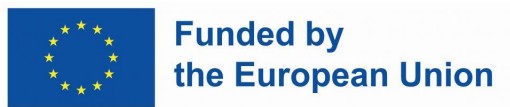

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

## A   FURTHER BACKGROUND & RELATED WORK

In the following, we outline further background information and related work on RNA folding dynamics.

The folding dynamics of RNA can be described as a stochastic process in a state space, comprised of a set of structures or conformations a given RNA sequence may assume, a move set that defines the allowed elementary transitions between conformations in the state space, and transition rates for all allowed transitions. Mathematically, this compiles into a continuous time Markov process governed by the following master equation for the state probabilities $P_x(t)$ of observing state $x$ at time $t$

$$\frac{dP_x(t)}{dt} = \sum_{y \neq x}[P_y(t)k_{xy} - P_x(t)k_{yx}]$$

where $k_{xy}$ is the transition rate from state $y$ to state $x$. For RNA sequences of moderate length, the master equation becomes too high dimensional to be solved analytically; therefore, it is approximated by Monte-Carlo simulation techniques (Flamm & Hofacker, 2008), which is, however, very time-consuming since enough stochastic simulations need to be accumulated to get a statistically representative time evolution of the state probabilities. Alternatively, acceleration has been proposed through a more macroscopic structural description of RNA by helix kinetics methods (Xayaphoummine et al., 2005; Danilova et al., 2006).

A different approach to simulating the dynamics of RNA folding is through analysis of the underlying folding landscape. Such a landscape can be constructed from complete suboptimal folding with *barriers* (Flamm et al., 2002), which provides an exact partitioning of the RNA conformation space into basins of attraction, i.e., local optima of the energy landscape. These macro-states provide a natural coarse-graining of the folding landscape and allow to re-formulate the dynamics on a reduced number of states, resulting in a massive speedup of computation time at comparable levels of detail. This idea is implemented in the tool *treekin*, which models the complete folding dynamics of RNA molecules of length up to approximately 100 nucleotides as a continuous-time Markov process that is solved by numerical integration (Wolfinger et al., 2004).

For molecular dynamics (MD) simulations, AI methods have already been applied in different parts of the MD pipeline. Deep learning methods like graph neural networks (GNNs) (Gilmer et al., 2017) or variational autoencoders (VAEs) (Kingma, 2013), as well as reinforcement learning (RL) (Sutton, 2018) are regularly used in these scenarios to e.g. enhance the sampling techniques during MD simulations, replace quantum mechanical force field simulations, or analyze the MD trajectories (Prašnikar et al., 2024). However, current approaches mainly focus on small molecule data due to the complexity of MD simulations for larger macromolecules and have the disadvantage that they require large amounts of simulation data for training (Prašnikar et al., 2024). For more information on AI-based methods in the field of MD simulations, we refer the interested reader to a detailed review of the field by Prašnikar et al. (2024).

## B  EXPONENTIAL *Kinfold* RUNTIME

Figure 7 shows the mean CPU times (in minutes), along with the upper bound standard deviations, for simulating 10, 25, 50, 75, and 1000 folding processes — transitioning from an open chain to the minimum free energy conformation — with the mean times calculated for different RNA sequence lengths based on 50 distinct artificial RNA molecules per length. Despite the logarithmic scale on the CPU time axis, the mean CPU time still shows a linear increase, highlighting the exponential growth in the computational time required for these simulations. The calculations for Figure 7 were performed on a single core of an AMD Milan EPYC 7513 CPU with 2.6 GHz.

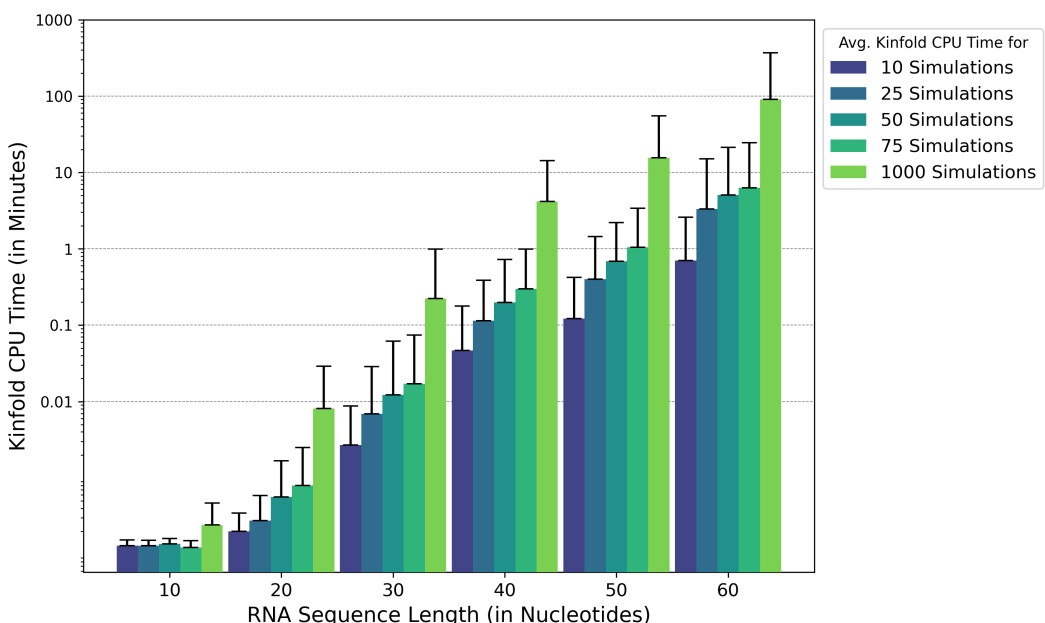

Figure 7: *Kinfold* mean CPU times (in minutes), including the upper bound standard deviations for simulating 10, 25, 50, 75, and 1000 folding processes over different RNA sequence lengths, based on 50 distinct artificial RNA molecules per length.

## C  SYNTHETIC FOLDING TIME DISTRIBUTION PRIOR DETAILS

In the following, we will describe our proposed synthetic prior and the method for sampling a single batch of synthetic first passage times from it in more detail. The synthetic first passage times $t$ are sampled from a distribution $p(\psi_k)$ generated from a family of multi-modal distributions $P_{\psi_k}$ as introduced in Section 4.1. The possible first passage time values across all $p(\psi_k)$ range from $10^\alpha$ to $10^\beta$, with $\alpha = -6$ and $\beta = 15$, thereby limiting $T \in [T_{\text{start}}, T_{\text{stop}}]$ by $\min(T_{\text{start}}) = 10^{-6}$ and $\max(T_{\text{stop}}) = 10^{15}$, as we observed that this time range covers a very high fraction of possible RNA folding processes.

Each distribution $p(\psi_k) \in P_{\psi_k}$ is characterized by $k$ Gaussian components, each with a mean $\mu_i$ and a standard deviation $\sigma_i$, for $i = 1, \ldots, k$. The base means $\mu_i^{\text{base}}$ are uniformly distributed between $\alpha + 1 = -5$ and $\beta + 1 = 16$, and the standard deviations $\sigma_i$ are uniformly distributed between $0.1$ and $\frac{\beta - \alpha}{5} = 4.2$. Further, we introduce a shifting parameter $\delta$, which is uniformly distributed between $\alpha$ and $\beta$, i.e., $\delta \sim \mathcal{U}(-6, 15)$ and is fixed for all $i = 1, \ldots, k$. The final means $\mu_i$ are then given by:

$$\mu_i = \mu_i^{\text{base}} + \delta.$$

Given the parameters $\psi_k$ and a value $x$, the probability density function (PDF) of the multi-modal Gaussian distribution is expressed as:

$$p(\psi_k, x) = \sum_{i=1}^{k} \exp\left( -\frac{(\log x - \mu_i)^2}{2\sigma_i^2} \right).$$

### C.1  SAMPLING FROM THE SYNTHETIC PRIOR OF RNA FIRST PASSAGE TIMES

To sample a batch of synthetic first passage times of size $B$ with a fixed number of times, i.e., number of simulations per training example of $M$ from a multi-modal distribution $p(\psi_k)$, we employ the inverse transformation method also known as the Smirnov transformation. To do so we generate the PDF $p(\psi_k, x)$ over a logarithmically spaced sequence $x$ of length $M$ from $10^\alpha$ to $10^\beta$. Then, to normalize this PDF and therefore ensure a valid probability distribution, we calculate:

$$\hat{p}(\psi_k, x) = \frac{p(\psi_k, x)}{\int_{10^\alpha}^{10^\beta} p'(\psi_k, \tau) \, d\tau}. \tag{6}$$

Next, we compute the cumulative distribution function (CDF):

$$\text{CDF}(\psi_k, x) = \int_{10^\alpha}^{x} \hat{p}(\psi_k, \tau) \, d\tau. \tag{7}$$

To ensure the CDF ranges from 0 to 1, we normalize it by dividing by the integral over the entire range from $10^\alpha$ to $10^\beta$:

$$\text{CDF}(\psi_k, x) = \frac{\int_{10^\alpha}^{x} \hat{p}(\psi_k, \tau) \, d\tau}{\int_{10^\alpha}^{10^\beta} \hat{p}(\psi_k, \tau) \, d\tau}. \tag{8}$$

This normalization ensures that the CDF is properly scaled, with $\text{CDF}(\psi_k, 10^\beta) = 1$.

By inverting the CDF, we obtain the quantile function $\text{CDF}^{-1}(\psi_k)$. To generate samples, we draw uniform samples $u_i$ from a uniform distribution $\mathcal{U}(0, 1)$ for $i = 1, \ldots, M$ and transform these samples using the inverse CDF:

$$t_i = \text{CDF}^{-1}(\psi_k, u_i),$$

where $t_i$ are the sampled values from the distribution. We then encode these samples by applying a logarithmic transformation:

$$\hat{t}_i = \log_{10}(t_i).$$

Finally, constructing the prior output, for a batch of size $B$ and a fixed number of first passage times per example $M$, we generate the independent variables $\mathbf{X}$ and $\mathbf{Y}$ as follows:

$$\mathbf{X} = \mathbf{0}_{B \times M \times 1},$$

$$\mathbf{Y}_{i,:} = [\hat{t}_1, \hat{t}_2, \ldots, \hat{t}_M] \quad \text{for } i = 1, \ldots, B.$$

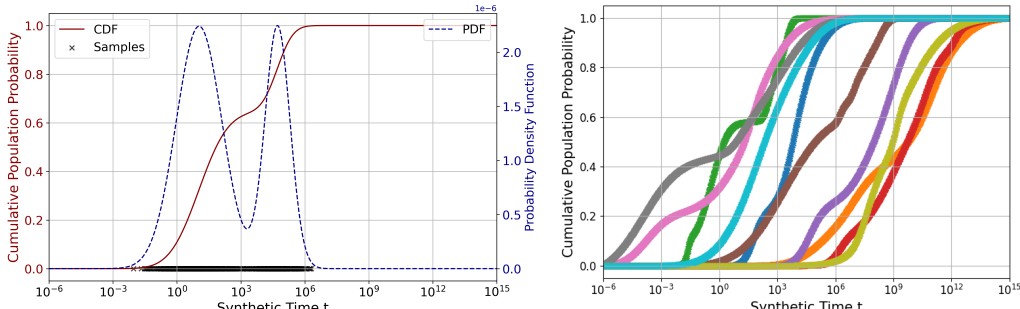

Figure 8: Examples of the synthetic prior of RNA first passage times. We show an example of a single CDF (red) and the corresponding multi-modal probability density function (PDF) (blue; dotted line) generated from the synthetic prior (left). The distribution is bi-modal ($k = 2$) with the parameters $\psi_k = \big((10.86, 1.36), (2.38, 2.48)\big)$. The right plot visualizes ten example CDFs generated from the synthetic prior.

# D  *KinPFN* DETAILS

## D.1  *KinPFN* HYPERPARAMETER

All hyperparameters in the *KinPFN* model are inherited from the transformer-based architecture (Vaswani et al., 2017) of prior-data fitted networks (PFNs) as proposed by Müller et al. (2022). These include the number of layers (*nlayers*), attention heads (*nheads*), embedding size (*emsize*), the number of neurons in each hidden layer (*nhidden*), the learning rate for the Adam optimizer (Kingma & Ba, 2015) (*learning_rate*), the number of steps per epoch (*steps*), and the total number of epochs (*epochs*). However, it is not entirely accurate to refer to "epochs" in this context, as we are training on synthetic data sampled from a prior, resulting in a single, infinite epoch. In the context of PFNs, the loss is updated after each step, which is why we describe these steps as hyperparameterized steps per epoch. The term "epochs" is used here primarily because it serves as a hyperparameter within the code, providing a mechanism to control the training process. Another crucial parameter is the sequence length (*seq_len*) of the input, representing the number of folding simulations (i.e., first passage times $M$) fed into the Transformer. This sequence length indicates the number of samples drawn from a prior distribution $p(\psi_k) \in P_{\psi_k}$, as defined in Section 4.1. Additionally, given the infinite nature of synthetic training data and the singular epoch, we set the dropout rate and the weight decay to zero.

## D.2  HYPERPARAMETER OPTIMIZATION

Given the uncertainty about the significance of each parameter in the final model's performance, we decided to utilize Neural Pipeline Search (NePS) (Stoll et al., 2023) for the hyperparameter optimization (HPO) of the *KinPFN* architecture. NePS is an open-source Python library that offers state-of-the-art HPO methods, including Bayesian Optimization and multi-fidelity methods like Hyperband (Li et al., 2017). In our setup, we chose Hyperband as our HPO technique. Hyperband optimizes the search process by dynamically allocating resources, enabling faster identification of the best configurations. It strikes an effective balance between exploration and exploitation. Initially, it explores a wide range of configurations with minimal resources, then progressively concentrates resources on the most promising candidates while discarding poor-performing ones early through a process of successive halving (Li et al., 2017).

As a performance metric for Hyperband to assess the quality of hyperparameter configurations, we utilize the prior-data negative log-likelihood (NLL), as outlined in Section 2. This approach is equivalent to calculating the Kullback-Leibler divergence between the approximated posterior predictive distribution (PPD) and the true target PPD (Müller et al., 2022). Each configuration trained by Hyperband is evaluated on a newly introduced validation set, discribed in Section G.

We conducted two final iterations of the NePS Hyperband process, evaluating a total of 261 configurations. After completing the first iteration, we made slight adjustments to the search space. Additionally, we set $N = 25$ for the validation pipeline in the first iteration and $N = 10$ for the second iteration, representing the number of context first passage times for each approximation. To ensure comparability across the validation of different hyperparameter configurations, we fixed the indices of these $N$ context first passage times within the available time points, which, in a real-world scenario, would typically be randomized since first passage times are usually obtained without any order when running kinetic folding algorithms like *Kinfold* (Flamm et al., 2000).

Table 2 and 3 outline the hyperparameter search space used for our optimization process in iteration one and two, respectively (differences are highlighted in blue). In the first iteration, we used a fixed batch size of 50. However, in the second iteration, we reduced the batch size to 40 to accommodate the adjusted search space, which brought us to our GPU memory limit. Since Hyperband requires a fidelity parameter to represent resource usage — in this case, computing time — we designate the *epochs* hyperparameter as the fidelity parameter, defining its range between 250 and 3000. This is directly related to the *steps* per epoch, as the model runs a specified number of steps during each epoch, with each step involving training on a single batch. By tuning both the number of epochs and steps per epoch, we control the amount of synthetic data sampled from the prior that our model sees during training. Additionally, we adjust the learning rate for the Adam optimizer (Kingma & Ba, 2015), setting a range between $10^{-5}$ and $10^{-3}$. This range is informed by preliminary training sessions, where we observed that higher learning rates resulted in highly irregular learning curves

Table 2: Hyperparameter search space for NePS Hyperband iteration 1.
Differences to iteration 2 are highlighted in blue

| Hyperparameter | Type | Values/Range |
|---|---|---|
| epochs | Integer | [250, 3000] (hyperband fidelity) |
| steps | Integer | [50, 100] |
| learning_rate | Float | $[10^{-5}, 10^{-3}]$ (log scale) |
| seq_len | Categorical | {200, 300, 500, 700} |
| buckets | Categorical | {100, 1000, 10000} |
| emsize | Categorical | {256, 512} |
| nheads | Categorical | {4, 8} |
| nhidden | Categorical | {512, 1024} |
| nlayers | Categorical | {2, 3, 4, 6, 8, 12} |

Table 3: Hyperparameter search space for NePS Hyperband iteration 2.
Differences to iteration 1 are highlighted in blue

| Hyperparameter | Type | Values/Range |
|---|---|---|
| epochs | Integer | [250, 3000] (hyperband fidelity) |
| steps | Integer | [50, 100] |
| learning_rate | Float | $[10^{-5}, 10^{-3}]$ (log scale) |
| seq_len | Categorical | {200, 300, 500, 700, 1000, 1400} |
| buckets | Categorical | {100, 1000, 5000, 10000} |
| emsize | Categorical | {256, 512} |
| nheads | Categorical | {4, 8} |
| nhidden | Categorical | {512, 1024} |
| nlayers | Categorical | {2, 3, 4, 6, 8} |

and, consequently, poor performance. We also evaluate models using different Transformer input sequence lengths — specifically 200, 300, 500, 700, 1000, and 1400 — as this parameter represents the number of first passage time samples $M$ drawn from each prior distribution $p(\psi_k)$. Furthermore, we assess the models with 100, 1000, 5000, and 10000 buckets over which we discretize the learned posterior predictive distribution. For the embedding size, we evaluate options of 256 and 512, and we asses 4 and 8 Transformer attention heads, which split the embedded input into smaller segments for focused attention. We also explore various model complexities by varying the number of neurons per hidden layer (512 and 1024) and the total number of layers, considering a broad range from 2 to 12 layers.

After both Hyperband iterations we identified four highly promising *KinPFN* architectures {$KinPFN_1, \dots KinPFN_4$}. Among these, *KinPFN*$_1$ and *KinPFN*$_3$ demonstrated the minimal NLL in the first and second NePS Hyperband iteration with 1.1761 ($N = 25$) and 1.2101 ($N = 10$), respectively. Table 4 shows the NLL performance metrics of the found configurations across various cutoffs $N \in \{10, 25, 50, 75, 100\}$. For each distribution example in the proposed validation set, we randomly selected the $N$ context times from the pool of $M = 1000$ available times, ensuring a broader and more generalizable evaluation, as the Hyperband validation pipeline was only based on fixed $N$ first passage times with fixed indices within $M$.

While *KinPFN*$_4$ was the configuration with the second-best mean NLL loss with 1.2102 ($N = 10$) after *KinPFN*$_3$ in the second NePS iteration, *KinPFN*$_2$ adopted the configuration of *KinPFN*$_1$ but trained on a larger Transformer input sequence length of 1400.

In the model analysis, *KinPFN*$_1$ shows the best performance with $N = 10$ context first passage times. However, for all other values of $N$ ($N \in \{25, 50, 75, 100\}$), *KinPFN*$_2$ surpasses it. Additionally, *KinPFN*$_2$ outperforms both models from the second NePS iteration, *KinPFN*$_3$ and *KinPFN*$_4$, based on the NLL losses, as demonstrated in Table 4. Based on these results, we selected *KinPFN*$_2$ as our final *KinPFN* model that was utilized in all experiments, as it shows the best overall performance.

Table 4: Comparison of four promising *KinPFN* hyperparameter configurations identified in two NePS (Stoll et al., 2023), i.e., Hyperband (Li et al., 2017) iterations in terms of prior-data negative log-likelihood loss (lower is better) with context first passage time cutoffs $N \in \{10, 25, 50, 75, 100\}$.

| Configuration | Parameters | First Passage Times $N$ | | | | |
|---|---|---|---|---|---|---|
| | | **10** | **25** | **50** | **75** | **100** |
| *KinPFN*$_1$ | seq_len=700, epochs=1000, steps=86, learning_rate=2.5588748050825984 $\times$ $10^{-5}$, buckets=1000, emsize=256, nheads=4, nhidden=512, nlayers=8, batch_size=50 | **1.348** | 1.254 | 1.225 | 1.216 | 1.210 |
| *KinPFN*$_2$ | seq_len=1400, epochs=1000, steps=86, learning_rate=2.5588748050825984 $\times$ $10^{-5}$, buckets=1000, emsize=256, nheads=4, nhidden=512, nlayers=8, batch_size=50 | 1.378 | **1.246** | **1.207** | **1.195** | **1.189** |
| *KinPFN*$_3$ | seq_len=1400, epochs=1000, steps=72, learning_rate=3.867480144966054 $\times$ $10^{-5}$, buckets=10000, emsize=256, nheads=4, nhidden=1024, nlayers=4, batch_size=40 | 1.384 | 1.255 | 1.219 | 1.208 | 1.202 |
| *KinPFN*$_4$ | seq_len=1400, epochs=333, steps=85, learning_rate=7.062252166123585 $\times$ $10^{-4}$, buckets=10000, emsize=512, nheads=4, nhidden=1024, nlayers=2, batch_size=40 | 1.418 | 1.259 | 1.215 | 1.202 | 1.194 |

**Final *KinPFN* Configuration**  The final *KinPFN* model consists of 4.86 million parameters, featuring a total of 8 layers, each with a hidden size of 512, 4 attention heads, an embedding size of 256, a learning rate of $2.5588748050825984 \times 10^{-5}$, and 1000 buckets. The model was trained for 1000 epochs, each consisting of 86 steps (with a batch size of 50), resulting in a total of 4,300,000 seen examples (calculated as 1000 x 86 x 50). Each example comprised $M = 1400$ (synthetic) first passage times from (theoretical) folding simulations, which represent the Transformer input sequence length.

# E   KDE AND DP-GMM DETAILS

To ensure an optimal comparison of *KinPFN* with Kernel Density Estimation (KDE) and the Dirichlet Process Gaussian Mixture Model (DP-GMM), we performed a random search hyperparameter optimization (HPO). For KDE, we tuned the *bandwidth* hyperparameter over a logarithmic search space ranging from $10^{-3}$ to $10^1$, while for DP-GMM, we optimized the *weight concentration prior* within a logarithmic range of $10^{-4}$ to $10^2$. Both methods were evaluated using 1,000 configurations, selecting the one with the lowest mean negative log-likelihood on our validation set, consisting of 2,019 real RNA first passage time distributions (Appendix G) using 25 context times for each example. Figure 9 illustrates the HPO results for KDE (left) and DP-GMM (right). As a result of the HPO, we selected a *bandwidth* of 0.352 with a Gaussian kernel for KDE and a *weight concentration prior* of 9.79e-4 for DP-GMM and allowed a maximum of 100,000 iterations of Expectation-Maximization (EM).

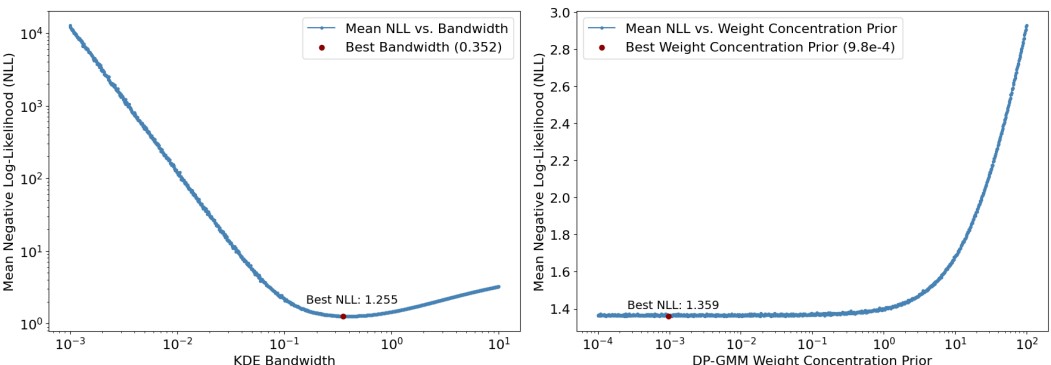

Figure 9: Hyperparameter optimization for Kernel Density Estimation (KDE) on the *bandwidth* parameter (left) and for Dirichlet Process Gaussian Mixture Models (DP-GMM) on the *weight concentration prior* parameter (right).

# F METRICS

In our experiments and evaluations, we rely on the prior-data negative log-likelihood (NLL) between the approximated posterior predictive distribution (PPD) and the true first passage time distribution as a primary performance metric, consistent with its use during training and hyperparameter optimization (HPO) (Section 4.2):

$$\ell_\theta = \mathbb{E}_{(0, t_{\text{test}}) \cup D_{\text{train}} \sim p(\psi_k)} \left[ -\log q_\theta(t_{\text{test}} | 0_{\text{test}}, D_{\text{train}}) \right] \qquad . \tag{9}$$

When comparing *KinPFN* to other methods, such as Gaussian Mixture Models (GMM), Dirichlet Process Gaussian Mixture Models (DP-GMM), and Kernel Density Estimation (KDE), we consistently use the mean negative log-likelihood (NLL) as the evaluation metric. This choice is motivated by the fact that the mean NLL reflects how effectively each method has learned the underlying posterior predictive distributions (PPDs) of the first passage times. Minimizing the NLL aligns with minimizing the Kullback-Leibler (KL) divergence between the estimated PPD and the ground truth PPD (Müller et al., 2022), making it a robust measure of model performance.

As our main objective is approximating the CDFs of the first passage times, we additionally evaluate the performance by measuring the mean absolute error (MAE) and Kolmogorov-Smirnov (KS) statistic between the CDF of the approximated PPD $\hat{F}(t)$ and the true target CDF $F(t)$. For a single CDF approximation of *KinPFN*, the mean absolute error (MAE) is defined as the average of the absolute differences between the predicted CDF values and the ground truth CDF values for a specific sequence of folding times. Mathematically, it can be expressed as:

$$\text{MAE} = \frac{1}{M} \sum_{i=1}^{M} \left| \hat{F}(t_i) - F(t_i) \right| \qquad , \tag{10}$$

where $M$ is the number of available ground truth first passage time points for the particular example RNA sequence, $\hat{F}(t_i)$ is the predicted CDF value at the $i$-th first passage time $t_i$, computed by *KinPFN*, $F(t_i)$ is the ground truth CDF value at the $i$-th first passage time $t_i$.

Similarly, the KS statistic is calculated as the maximum absolute difference between the predicted and true CDFs:

$$\text{KS Statistic} = \max_{t_i} \left| \hat{F}(t_i) - F(t_i) \right| \qquad ,$$

where lower values indicate a better fit of the model to the true distribution.

## G    VALIDATION AND TEST DATA

We introduce two new datasets: a validation set and a test set, both consisting of real RNA first passage times. The validation set contains 2,016 randomly generated RNA sequences, while the test set includes 635 sequences. The times were acquired by simulating the folding process of the RNAs, starting from an open-chain conformation and progressing to the molecule's minimum free energy conformation with the kinetic folding simulator *Kinfold* (Flamm et al., 2000). Figure 10 illustrates the distribution of RNA sequence lengths across both datasets. The validation set, used throughout all NePS (Stoll et al., 2023) iterations (i.e., Hyperband (Li et al., 2017)), is shown in dark blue, while the test set, shown in dark red, is reserved for final *KinPFN* model evaluations (see Section 5.1). Importantly, these two datasets are mutually independent in terms of RNA primary sequences and secondary structures.

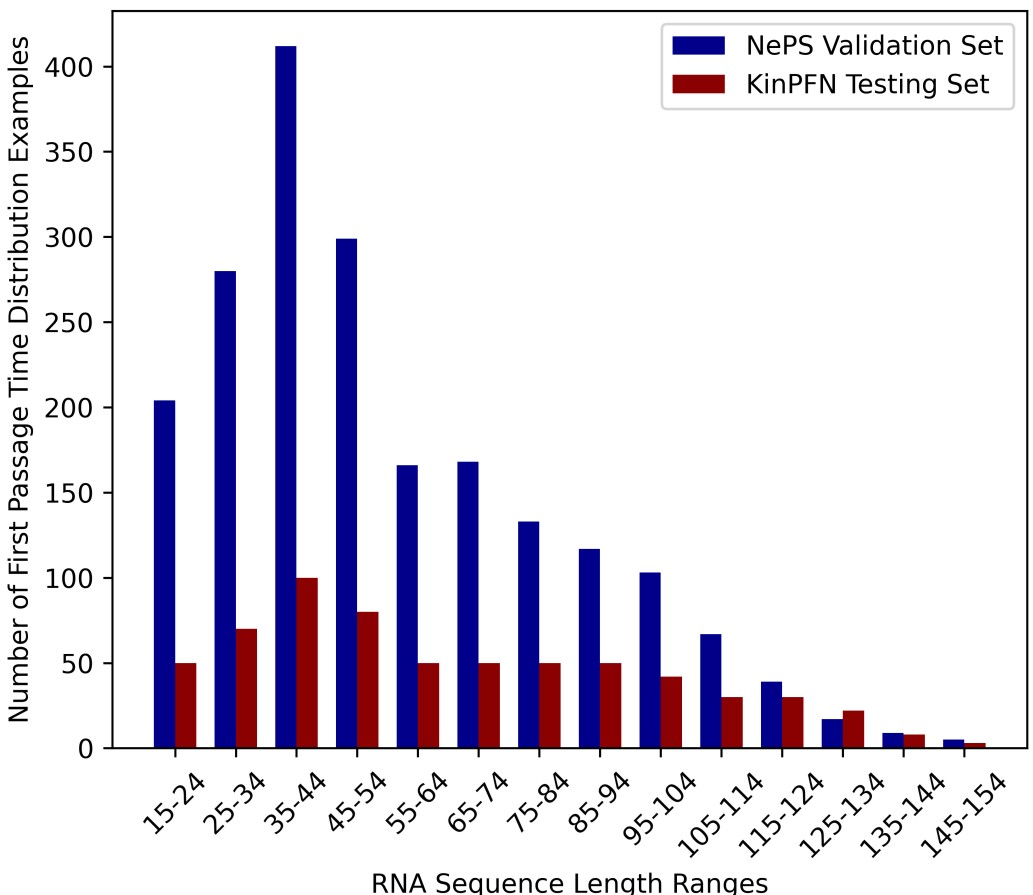

Figure 10: Number of examples by RNA sequence length ranges for the custom validation set used in all NePS (Stoll et al., 2023) i.e., Hyperband (Li et al., 2017) iterations and the custom testing set used for final *KinPFN* model evaluations (Section 5.1). Both sets are independent of each other with respect to RNA primary sequence and secondary structures.

# H    ADDITIONAL EVALUATIONS

## H.1    SYNTHETIC PRIOR APPROXIMATIONS

We evaluate our proposed model using synthetic data generated from the same prior distribution employed during training (see Section 4.1). We approximate 10,000 synthetic first passage time distributions, varying the cutoff points for the number of context first passage times $N \in \{10, 25, 50, 75, 100\}$. This allows us to evaluate the model's performance as the number of context points provided to *KinPFN* increases. For each case, we sample $M = 1000$ first passage times from the prior distribution. The performance is measured in terms of the posterior predictive distribution (PPD) mean negative log-likelihood (NLL) and the cumulative distribution function (CDF) mean absolute error (MAE), computed over all 10,000 examples at each cutoff $N$. Table 5 presents the results of this evaluation. We observe significant improvements in both the NLL and MAE when increasing the number of context points from $N = 10$ to $N = 25$ and from $N = 25$ to $N = 50$. Beyond $N = 50$, while the loss continues to decrease, the rate of improvement slows down as the context size grows from $N = 75$ to $N = 100$.

Table 5: Performance evaluation of *KinPFN* on 10,000 synthetic first passage time distributions. Metrics are shown for different context first passage time cutoffs $N \in \{10, 25, 50, 75, 100\}$, measured in terms of negative log-likelihood (NLL) and mean absolute error (MAE). Lower values indicate better performance.

| Performance Metric | First Passage Times $N$ | | | | |
|---|---|---|---|---|---|
| | 10 | 25 | 50 | 75 | 100 |
| Mean Prior-Data NLL | 2.4265 | 2.1364 | 2.0596 | 2.0388 | 2.0281 |
| Mean Absolute Error | 0.0878 | 0.0553 | 0.0388 | 0.0321 | 0.0275 |

## H.2    COMPARISON *KinPFN*, *GMM*, *DP-GMM* AND *KDE*

To further evaluate our model, we compare *KinPFN* against multiple Gaussian Mixture Models (GMMs) and Dirichlet Process Gaussian Mixture Models (DP-GMMs) using various initial modality assumptions. Specifically, we consider mixture models with modalities $k \in \{2, 3, 4, 5\}$, aligning with the assumptions outlined in our synthetic prior (Section 4.1), denoted as $GMM_k$ and $DP\text{-}GMM_k$. For all evaluations, the models were provided identical context first passage times. Both GMM and DP-GMM models were allowed a maximum of 100,000 iterations of Expectation-Maximization (EM). Additionally, we compare *KinPFN* to a Kernel Density Estimator (KDE) that we optimized for its bandwidth hyperparameter (Appendix E), as discussed in Section 5.1. The results for MAE and KS are shown in Table 6 and 7, respectively. We observe that *KinPFN* outperforms all other methods from context size of 25 FPTs onwards. Results on larger context sizes are shown in Table 8, 9 and 10 demonstrating a constant improvement of the predictions with growing context sizes. Finally, we also compare *KinPFN* with GMM ensembles with modalities $k = \{2, 3, 4, 5\}$ and $k = \{2, 3, 4\}$, weighted according to their marginal likelihoods, across different context sizes. The results are shown in Table 11 and 12 for NLL and MAE, respectively.

Table 6: Evaluation of *KinPFN*, *KDE*, and multiple $GMM_k$ and $DP\text{-}GMM_k$ models with different initial modality assumptions $k \in \{2, 3, 4, 5\}$ on a newly introduced testing set comprising 635 real-world first passage time distributions in terms of mean absolute error (lower is better) with context first passage time cutoffs $N \in \{10, 25, 50, 75, 100\}$.

| Method | First Passage Times $N$ | | | | |
|---|---|---|---|---|---|
| | 10 | 25 | 50 | 75 | 100 |
| *KinPFN* | 0.0843 | **0.0561** | **0.0393** | **0.0333** | **0.0296** |
| $GMM_2$ | 0.1003 | 0.0848 | 0.0790 | 0.0773 | 0.0756 |
| $GMM_3$ | 0.0988 | 0.0866 | 0.0815 | 0.0801 | 0.0778 |
| $GMM_4$ | 0.0952 | 0.0860 | 0.0816 | 0.0799 | 0.0777 |
| $GMM_5$ | 0.0929 | 0.0842 | 0.0809 | 0.0797 | 0.0778 |
| $DP\text{-}GMM_2$ | 0.0866 | 0.0774 | 0.0761 | 0.0756 | 0.0745 |
| $DP\text{-}GMM_3$ | 0.0867 | 0.0774 | 0.0763 | 0.0762 | 0.0751 |
| $DP\text{-}GMM_4$ | 0.0865 | 0.0770 | 0.0763 | 0.0763 | 0.0751 |
| $DP\text{-}GMM_5$ | 0.0860 | 0.0768 | 0.0762 | 0.0760 | 0.0751 |
| *KDE* | **0.0813** | 0.0690 | 0.0653 | 0.0644 | 0.0630 |

Table 7: Evaluation of *KinPFN*, *KDE*, and multiple $GMM_k$ and $DP\text{-}GMM_k$ models with different initial modality assumptions $k \in \{2, 3, 4, 5\}$ on a newly introduced testing set comprising 635 real-world first passage time distributions in terms of Kolmogorov-Smirnov (KS) statistic (lower is better) with context first passage time cutoffs $N \in \{10, 25, 50, 75, 100\}$.

| Method | First Passage Times $N$ | | | | |
|---|---|---|---|---|---|
| | 10 | 25 | 50 | 75 | 100 |
| *KinPFN* | 0.1615 | **0.1098** | **0.0809** | **0.0700** | **0.0632** |
| $GMM_2$ | 0.2084 | 0.1705 | 0.1586 | 0.1541 | 0.1510 |
| $GMM_3$ | 0.2210 | 0.1794 | 0.1644 | 0.1586 | 0.1537 |
| $GMM_4$ | 0.2293 | 0.1829 | 0.1674 | 0.1604 | 0.1547 |
| $GMM_5$ | 0.2352 | 0.1836 | 0.1695 | 0.1625 | 0.1564 |
| $DP\text{-}GMM_2$ | 0.1695 | 0.1505 | 0.1496 | 0.1488 | 0.1471 |
| $DP\text{-}GMM_3$ | 0.1694 | 0.1506 | 0.1499 | 0.1495 | 0.1475 |
| $DP\text{-}GMM_4$ | 0.1691 | 0.1500 | 0.1500 | 0.1496 | 0.1475 |
| $DP\text{-}GMM_5$ | 0.1682 | 0.1494 | 0.1499 | 0.1491 | 0.1476 |
| *KDE* | **0.1590** | 0.1344 | 0.1278 | 0.1256 | 0.1231 |

Table 8: Evaluation of *KinPFN*, *KDE*, and multiple *GMM$_k$* and *DP-GMM$_k$* models with different initial modality assumptions $k \in \{2, 3, 4, 5\}$ on a newly introduced testing set comprising 635 real-world first passage time distributions in terms of prior-data negative log-likelihood loss (lower is better) with context first passage time cutoffs $N \in \{250, 500, 750, 1000\}$.

| Method | First Passage Times $N$ | | | |
|---|---|---|---|---|
| | 250 | 500 | 750 | 1000 |
| *KinPFN* | 1.1756 | 1.1716 | 1.1703 | 1.1697 |
| *GMM$_2$* | 1.1764 | 1.1715 | 1.1696 | 1.1690 |
| *GMM$_3$* | 1.1621 | 1.1542 | 1.1519 | 1.1508 |
| *GMM$_4$* | **1.1612** | 1.1506 | 1.1476 | 1.1458 |
| *GMM$_5$* | 1.1642 | **1.1499** | **1.1458** | **1.1438** |
| *DP-GMM$_2$* | 1.1853 | 1.1735 | 1.1700 | 1.1683 |
| *DP-GMM$_3$* | 1.1793 | 1.1627 | 1.1565 | 1.1533 |
| *DP-GMM$_4$* | 1.1802 | 1.1631 | 1.1562 | 1.1526 |
| *DP-GMM$_5$* | 1.1809 | 1.1637 | 1.1566 | 1.1529 |
| *KDE* | 1.1874 | 1.1841 | 1.1832 | 1.1828 |

Table 9: Evaluation of *KinPFN*, *KDE*, and multiple *GMM$_k$* and *DP-GMM$_k$* models with different initial modality assumptions $k \in \{2, 3, 4, 5\}$ on a newly introduced testing set comprising 635 real-world first passage time distributions in terms of mean absolute error (lower is better) with context first passage time cutoffs $N \in \{250, 500, 750, 1000\}$.

| Method | First Passage Times $N$ | | | |
|---|---|---|---|---|
| | 250 | 500 | 750 | 1000 |
| *KinPFN* | **0.0205** | **0.0155** | **0.0137** | **0.0126** |
| *GMM$_2$* | 0.0742 | 0.0730 | 0.0730 | 0.0728 |
| *GMM$_3$* | 0.0764 | 0.0756 | 0.0756 | 0.0754 |
| *GMM$_4$* | 0.0763 | 0.0754 | 0.0753 | 0.0751 |
| *GMM$_5$* | 0.0762 | 0.0751 | 0.0751 | 0.0747 |
| *DP-GMM$_2$* | 0.0746 | 0.0741 | 0.0739 | 0.0736 |
| *DP-GMM$_3$* | 0.0760 | 0.0757 | 0.0759 | 0.0757 |
| *DP-GMM$_4$* | 0.0759 | 0.0756 | 0.0758 | 0.0756 |
| *DP-GMM$_5$* | 0.0759 | 0.0756 | 0.0758 | 0.0756 |
| *KDE* | 0.0624 | 0.0617 | 0.0617 | 0.0615 |

Table 10: Evaluation of *KinPFN*, *KDE*, and multiple *GMM$_k$* and *DP-GMM$_k$* models with different initial modality assumptions $k \in \{2, 3, 4, 5\}$ on a newly introduced testing set comprising 635 real-world first passage time distributions in terms of Kolmogorov-Smirnov (KS) statistic (lower is better) with context first passage time cutoffs $N \in \{250, 500, 750, 1000\}$.

| Method | First Passage Times $N$ | | | |
|---|---|---|---|---|
| | 250 | 500 | 750 | 1000 |
| *KinPFN* | **0.0484** | **0.0389** | **0.0359** | **0.0336** |
| *GMM$_2$* | 0.1482 | 0.1465 | 0.1463 | 0.1460 |
| *GMM$_3$* | 0.1493 | 0.1466 | 0.1467 | 0.1462 |
| *GMM$_4$* | 0.1489 | 0.1462 | 0.1461 | 0.1454 |
| *GMM$_5$* | 0.1499 | 0.1464 | 0.1462 | 0.1454 |
| *DP-GMM$_2$* | 0.1474 | 0.1472 | 0.1469 | 0.1465 |
| *DP-GMM$_3$* | 0.1483 | 0.1473 | 0.1469 | 0.1463 |
| *DP-GMM$_4$* | 0.1481 | 0.1471 | 0.1466 | 0.1461 |
| *DP-GMM$_5$* | 0.1482 | 0.1470 | 0.1465 | 0.1460 |
| *KDE* | 0.1217 | 0.1205 | 0.1206 | 0.1204 |

Table 11: Evaluation of *KinPFN* and two *GMM-Ensembles* with different modality assumptions $k = \{2, 3, 4, 5\}$ and $k = \{2, 3, 4\}$, weighted according to their marginal likelihoods, on a newly introduced testing set comprising 635 real-world first passage time distributions in terms of prior-data negative log-likelihood loss (lower is better) with context first passage time cutoffs $N \in \{10, 25, 50, 75, 100\}$.

| Method | First Passage Times $N$ | | | | |
|---|---|---|---|---|---|
| | 10 | 25 | 50 | 75 | 100 |
| *KinPFN* | **1.374** | **1.244** | **1.205** | **1.192** | **1.186** |
| *GMM-Ensemble$_{2,3,4,5}$* | 6.417 | 1.917 | 1.389 | 1.261 | 1.218 |
| *GMM-Ensemble$_{2,3,4}$* | 4.270 | 1.639 | 1.312 | 1.228 | 1.202 |

Table 12: Evaluation of *KinPFN* and two *GMM-Ensembles* with different modality assumptions $k = \{2, 3, 4, 5\}$ and $k = \{2, 3, 4\}$, weighted according to their marginal likelihoods, on a newly introduced testing set comprising 635 real-world first passage time distributions in terms of mean absolute error (lower is better) with context first passage time cutoffs $N \in \{10, 25, 50, 75, 100\}$.

| Method | First Passage Times $N$ | | | | |
|---|---|---|---|---|---|
| | 10 | 25 | 50 | 75 | 100 |
| *KinPFN* | **0.084** | **0.056** | **0.039** | **0.0333** | **0.030** |
| *GMM-Ensemble$_{2,3,4,5}$* | 0.093 | 0.084 | 0.081 | 0.0800 | 0.078 |
| *GMM-Ensemble$_{2,3,4}$* | 0.095 | 0.086 | 0.081 | 0.0800 | 0.078 |

### H.3  ADDITIONAL *KinPFN* APPROXIMATION EXAMPLES

In Section 5.1, we conducted approximations of first passage time distributions using *KinPFN* on a 75-nucleotide RNA. The folding process for this RNA was simulated with the *Kinfold* kinetic simulator (Flamm et al., 2000), employing custom start and stop structures. We also approximated the folding time distribution for a 56-nucleotide RNA, for which we obtained ground truth data using the *KFold* simulator (Dykeman, 2015). In all instances, the approximations were based on just 25 context first passage times. This appendix provides additional approximations for these RNAs, expanding the analysis by varying the number of context first passage times, specifically $N \in \{10, 25, 50, 75\}$. Additionally, we performed approximations for a 93-nucleotide RNA using the same simulation method as for the 75-nucleotide RNA (Figure 11) and extended our analysis of the 56-nucleotide RNA to include results for a 31-nucleotide RNA (Figure 12). Additionally, Figure 13 presents representative approximations for two RNAs from our newly introduced test set: a 97-nucleotide RNA and a 119-nucleotide RNA, each evaluated with four different numbers of context first passage times, $N \in \{10, 25, 50, 75\}$.

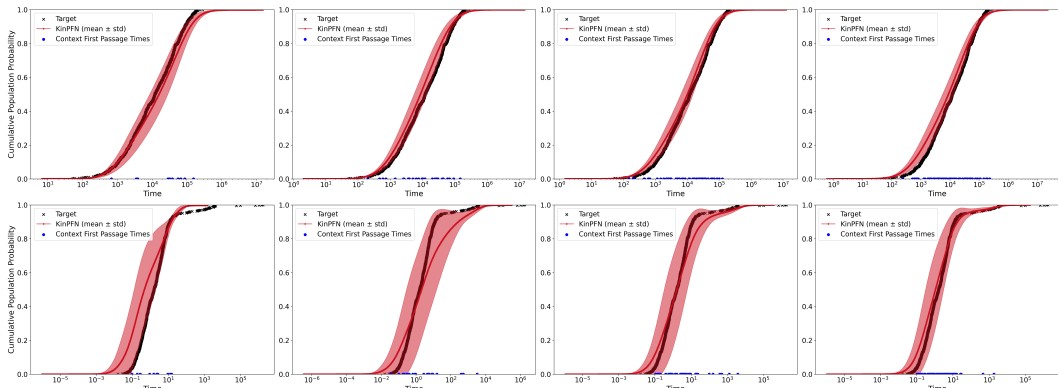

Figure 11: Approximations of the cumulative distribution function (CDF) for the first passage time using *KinPFN*, with context times $N \in \{10, 25, 50, 75\}$ (left to right), for an RNA sequence of 75 nucleotides (top) and 93 nucleotides (bottom). The folding process was simulated using custom initial and final structures rather than the open chain and minimum free energy conformation.

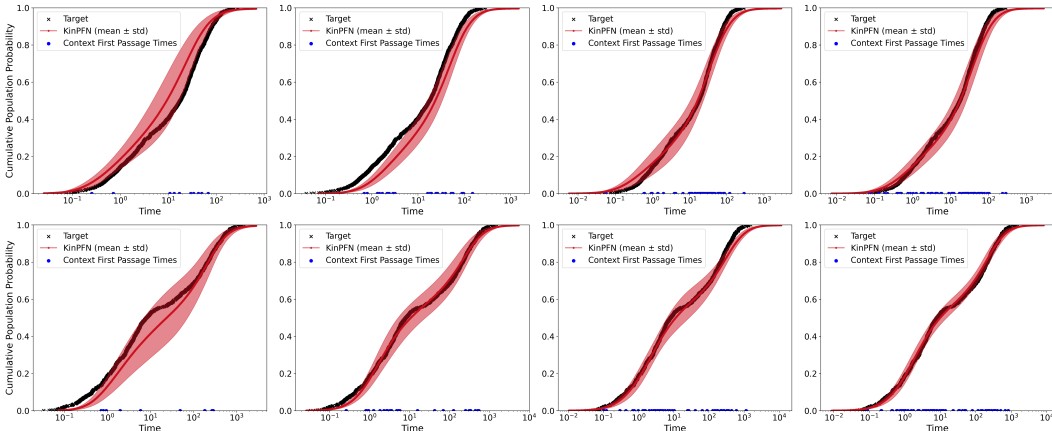

Figure 12: *KinPFN* first passage time CDF approximations with context times $N \in \{10, 25, 50, 75\}$ (left to right) for a 31 nucleotide long RNA (top) and a 56 nucleotide long RNA (bottom). First passage times were obtained using the kinetic folding algorithm *Kfold* Dykeman (2015).

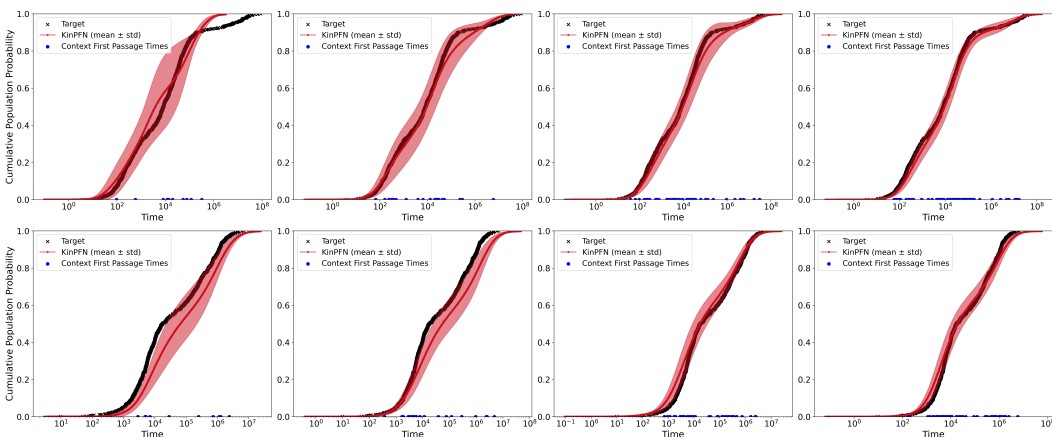

Figure 13: *KinPFN* first passage time CDF approximations with context times $N \in \{10, 25, 50, 75\}$ (left to right) for a 97 nucleotide long RNA (top) and a 119 nucleotide long RNA (bottom) that are part of the newly introduced test set.

## H.4 *KinPFN* Riemann Distribution Visualization

To provide a more intuitive understanding of the actual *KinPFN* predictions, Figure 14 illustrates two CDF approximations for different RNA molecules, each with $N = 25$ context first-passage times, compared against the ground truth CDF based on 1,000 *Kinfold* (Flamm et al., 2000) times (left) alongside the probability density function (PDF) of the corresponding approximated posterior predictive distribution (PPD), also known as the Riemann distribution (right) (Müller et al., 2022). As discussed in Section 4.2, the PFN predicts a continuous distribution, which we discretized into a finite number of buckets, forming the PDF bars. Each bar represents a bucket, and in our final *KinPFN* model, we used 1,000 buckets, a hyperparameter defined in Section 4.2.

By examining the approximated PPD PDFs, we can observe the multi-modal nature of the learned first-passage time distributions. For instance, the distribution for a 65 nucleotide long RNA in Figure 14 (top) shows bi-modality with two distinct peaks, while another first passage time distribution for a 86 nucleotide long RNA in Figure 14 (bottom) exhibits tri-modality with three peaks. Notably, the calculated CDFs for these multi-modal PPDs align closely with the ground truth CDFs from the 1,000 real first-passage times, demonstrating the effectiveness of our proposed prior, specifically, the family of multi-modal Gaussian distributions introduced in Section 4.1. This prior, from which we sampled synthetic first-passage time distributions to train *KinPFN*, enabled a strong generalization to real-world RNA first-passage time distributions.

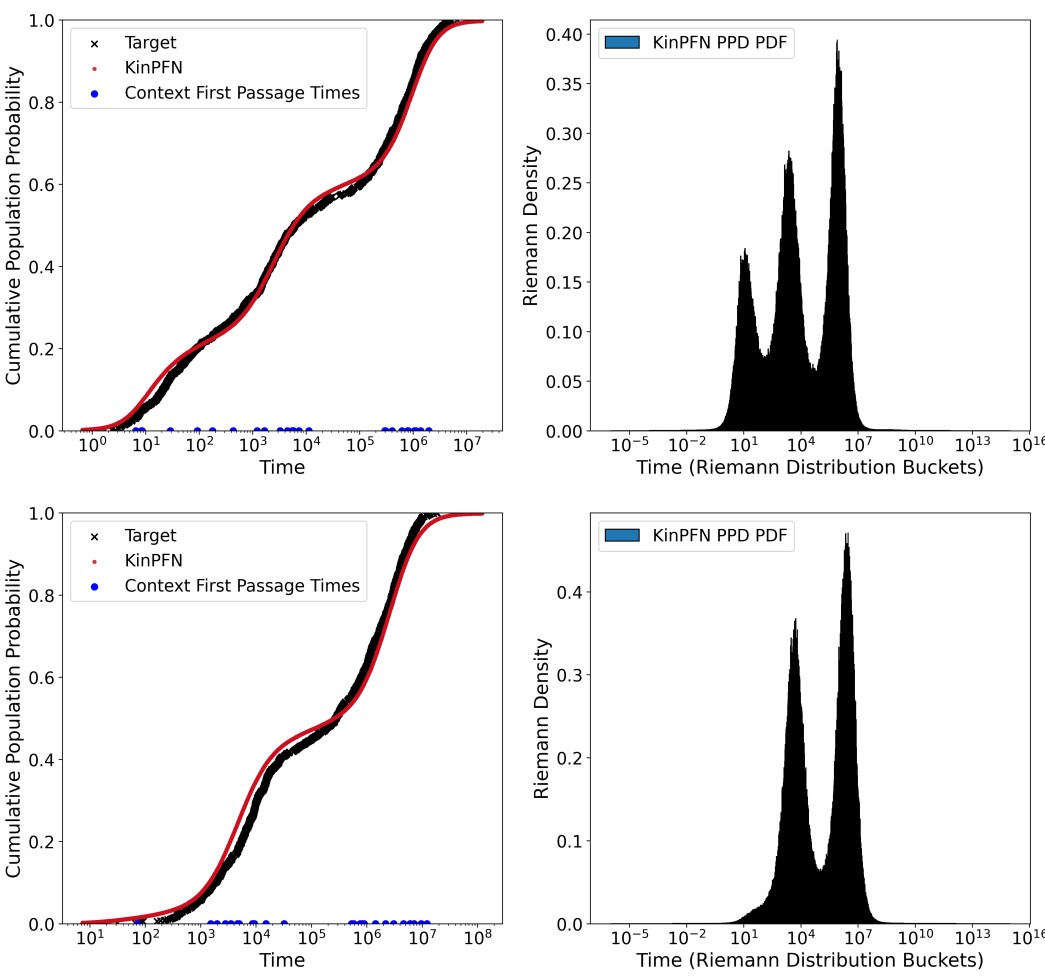

Figure 14: *KinPFN* PPD CDF approximations (left) along with the corresponding multi-modal PPD PDFs (right) for two RNA molecules with lengths of 65 (top) and 86 (bottom) nucleotides.

## H.5 EUKARYOTIC TRANSFER AND RIBOSOMAL RNA

We present additional results for the first passage time distribution approximations using *KinPFN* for tRNA$^{phe}$ and 5S rRNA from the eukaryote *Saccharomyces cerevisiae*, also known as brewer's yeast. Figure 15 displays the cumulative distribution function (CDF) approximations of first passage times for tRNA$^{phe}$ (top) and 5S rRNA (bottom), with varying numbers of context first passage times, $N \in \{10, 25, 50, 75\}$, as inputs to *KinPFN*. Additionally, Figure 16 illustrates the CPU time required (in minutes) to compute 10, 25, 50, 75, and 1,000 first passage times for both tRNA$^{phe}$ (blue) and 5S rRNA (red) using the kinetic simulator *Kinfold* (Flamm et al., 2000).

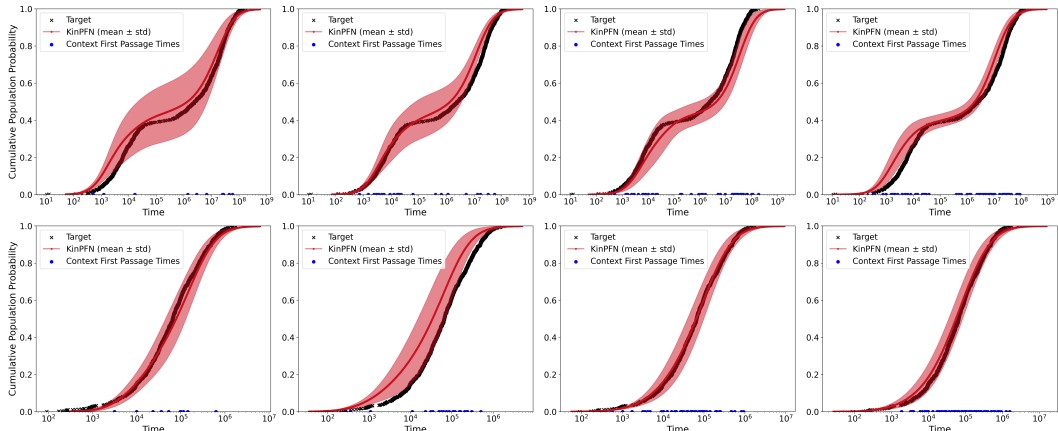

Figure 15: *KinPFN* first passage time CDF approximations with context times $N \in \{10, 25, 50, 75\}$ (left to right) for *Saccharomyces cerevisiae* tRNA$^{phe}$ (top) and 5S rRNA (bottom).

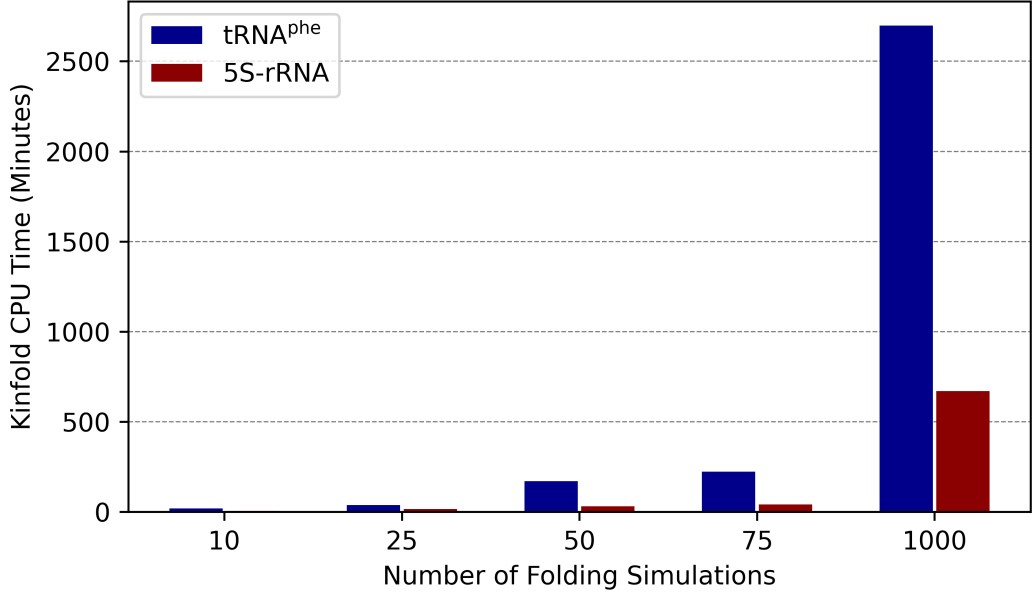

Figure 16: *Kinfold* CPU time (in minutes) vs. number of folding simulations for *Saccharomyces cerevisiae* tRNA$^{phe}$ and 5S rRNA.

### H.6 APPROXIMATIONS FOR ADDITIONAL RNA TYPES

In this section, we show the results of the first passage time distribution approximations using *KinPFN* on two further RNA types: hsa-miR-7107-3p (RNAcentral-ID: URS0000759FB2_9606) from *Homo sapiens* (human) and the SAM riboswitch (S box leader) (RNAcentral-ID: URS00002F3927_224308) from *Bacillus subtilis subsp. subtilis str. 168*. Figure 17 displays the cumulative distribution function (CDF) approximations for the first passage times of these RNAs. The top row shows results for hsa-miR-7107-3p, a 27-nucleotide microRNA, while the bottom row illustrates results for the SAM riboswitch, a 92-nucleotide regulatory RNA. For both RNAs, the approximations are generated using varying numbers of context first passage times as inputs to *KinPFN*, with $N \in \{10, 25, 50, 75\}$ (from left to right). These results provide additional insights into the robustness and versatility of *KinPFN* across diverse RNA types.

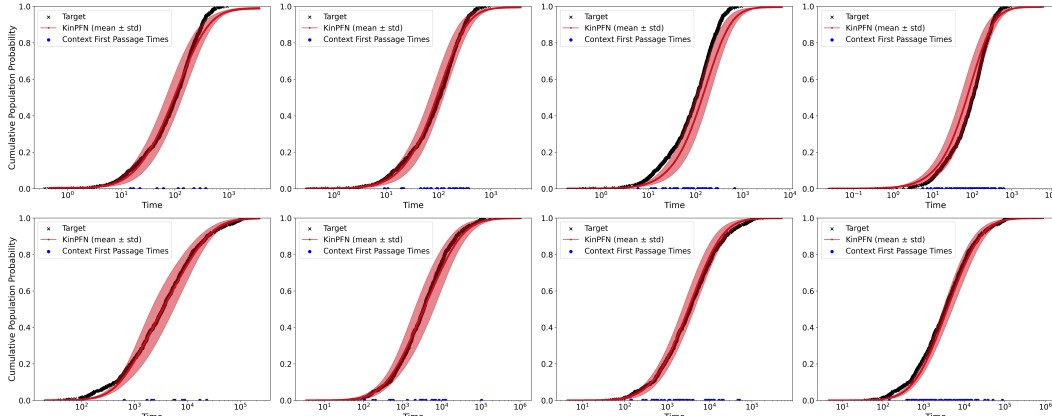

Figure 17: *KinPFN* first passage time CDF approximations with context times $N \in \{10, 25, 50, 75\}$ (left to right) for a hsa-miR-7107-3p microRNA from *Homo sapiens* (human) (top) and a SAM riboswitch from *Bacillus subtilis subsp. subtilis str. 168* (bottom).

## H.7 APPLICATION: RNA FOLDING EFFICIENCY ANALYSIS

Here we show *KinPFN* approximations using additional context first passage times of $N \in \{10, 25, 50\}$ on a case study that focuses on comparing the folding efficiency of three 43-nucleotide RNA molecules ($\phi_0$, $\phi_1$, $\phi_2$), each predicted to fold into the same minimum free energy (MFE) structure ($\omega_{\text{stop}} = $ .........................(((((....)))))....). As noted in Section 5.3, *KinPFN* accurately captures the overall folding behavior of these RNAs using just ten context times. We further observe that increasing the number of context first passage times to 25 and 50 enhances the accuracy of these approximations, as shown in Figure 18.

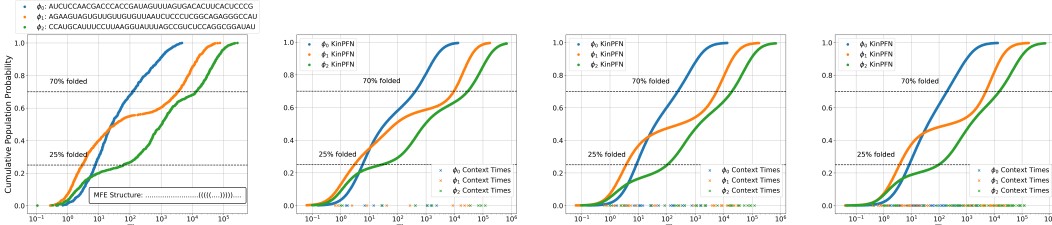

Figure 18: RNA folding efficiency analysis. The first plot from the left shows the ground truth CDFs $F(t)$ for three sequences $\phi_0, \phi_1$ and $\phi_2$, representing the fraction of molecules folded into the MFE conformation over time $t$. The second, third, and fourth plot displays the *KinPFN* approximations $\hat{F}(t)$ with ten, 25, and 50 *Kinfold* times as context.

## H.8 *KinPFN* Generalizes to Gene Expression Data

In this section, we present supplementary approximations of mRNA gene expression data for interleukin-1-$\alpha$ (*IL-1$\alpha$*), interleukin-1-$\beta$ (*IL-1$\beta$*), and tumor necrosis factor-alpha (*TNF-$\alpha$*). These additional results build upon the experiment described in Section 5.4, where *KinPFN* was applied to predict cumulative probability functions for gene expression in RAW 264.7 and BMDM cells using a subset of context data points. The approximations shown here utilize further numbers of context data points (10, 25, 50 and 75) and a different seed, providing a more comprehensive understanding of the predictive capabilities of *KinPFN* across different biological datasets. The results in this section serve to reinforce the main findings and demonstrate the robustness of *KinPFN* in accurately replicating the outcomes of experiments such as those conducted by Bagnall et al. (2020) while using minimal input data.

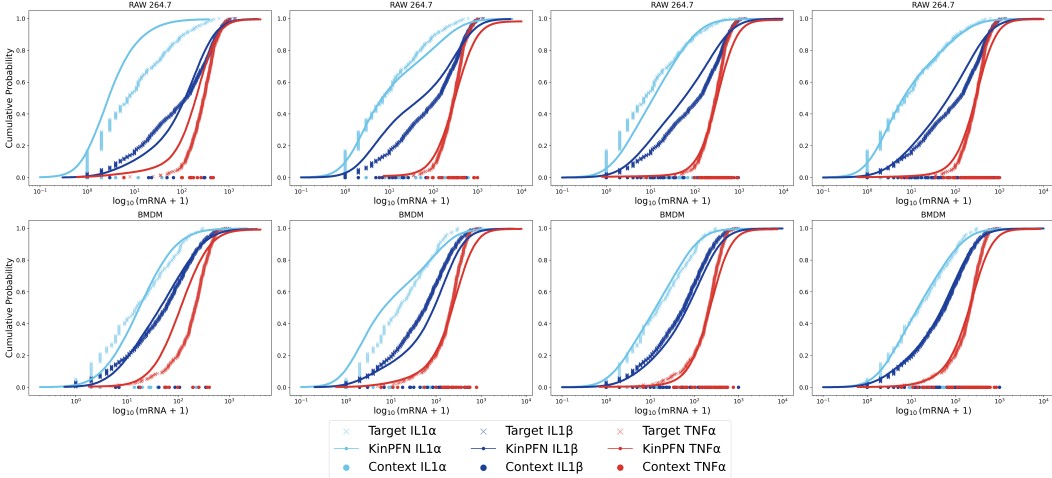

Figure 19: Approximation of mRNA expression of *IL-1$\alpha$*, *IL-1$\beta$* and *TNF-$\alpha$* in RAW 264.7 and BMDM cells. We plot approximations using only 10, 25, 50 and 75 context data points per gene.

