# OpenReview forum: "KinPFN: Bayesian Approximation of RNA Folding Kinetics using Prior-Data Fitted Networks"
_ICLR.cc/2025/Conference — ICLR 2025 Poster_

### Official Review · Reviewer_FZT1 · 2024-11-03

**Soundness:** 1
**Presentation:** 3
**Contribution:** 1
**Rating:** 3
**Confidence:** 4

**Summary:**

Biomolecules adopt many conformation in situ and understanding the kinetics of the transitions between these conformations is useful for understanding their biophysical behavior. Expensive simulations can start at one conformation and measure how long it takes to transition to another conformation -- the passage times. To minimize the number of simulations needed to characterize the distribution of first passage times, the authors build a prior on the distribution of first passage times; this in principle allows them to do more efficient inference with fewer measurements.

The authors focus on RNA kinetics and expression data. They build their prior by fitting a language model on sequences of passage times. They demonstrate they fit data better than Dirichlet process and kernel density estimator baselines and have reasonable fits on real data.

**Strengths:**

The paper is sound, and has extensive experimental validation.

**Weaknesses:**

See questions.

**Questions:**

1. Why not train on real passage time data that you can simulate? Why synthetic?

2. The prior described by the synthetic data is so simple it may be easy to just run MCMC. Why use a language model at all?

3. Language models as priors can have some pathologies. How does the model behave as the amount of data becomes large? In figure 11 it looks like even with a lot of data the PFN doesn't converge to the true CDF.

---

> ### Author Response · Authors · 2024-11-21
> **Response to Reviewer FZT1**
>
> Dear Reviewer FZT1,
>
> Thank you for your positive feedback and highlighting the soundness of our approach.
>
> >Why not train on real passage time data that you can simulate? Why synthetic?
>
> Generating a sufficiently large dataset for training a deep learning approach is currently infeasible due to the exponential increase in computation time required by Kinfold (and other kinetic simulators) as the RNA sequence length grows linearly. This is caused by an exponential growth of the conformational space S_n with the sequence length n, roughly following S_n \approx 1.86^n. Additionally, a dataset consisting of simulation data from a single simulator might be strongly biased, and the trained deep learning approach would potentially struggle to generalize across different simulators. However, our initial approach involved training a PFN using real first-passage time data that we generated by running Kinfold on a large set of very short sequences. However, training a PFN on real simulation data resulted in substantially worse performance compared to KinPFN, even when evaluating on data obtained from the same simulator.
>
> Generally, the generation of synthetic data for biological applications is challenging because we cannot make any assumptions about the underlying data generating process. We, therefore, decided to approach the problem by directly learning the distribution of first passage times in an in-context learning setup that allows us to condition on previous observations (e.g. obtained from simulator data). Besides providing improved performance, this approach makes KinPFN broadly applicable to RNA folding kinetics independent of the sequence length, the simulator used for generating the context data, the RNA type, the energy differences between the start and stop structure, and even the specific problem addressed by the approximations (as shown by our generalization to gene expression data).
>
> We think that training on synthetic data could solve many current issues connected to the usage of deep learning methods, particularly in the life sciences. These issues e.g. include overfitting due to limited amounts of available training data,unbalanced training data, as well as fairness problems. With KinPFN, we demonstrate that even with a relatively simple synthetic prior, we could generate synthetic data that represents a biological problem rather well, leading to strong approximation quality, avoiding the risk of overfitting, while learning from unlimited amounts of data that provides full control over data parametrizations.
>
> >The prior described by the synthetic data is so simple it may be easy to just run MCMC. Why use a language model at all?
>
> Monte Carlo methods are regularly used in RNA folding kinetics and we agree that it would be beneficial to exclusively train on MCMC-generated data. However, the runtime of MCMC depends on the chain length, and thus on the number of structural states. Therefore, running MCMC on long RNAs is computationally infeasible.
>
> Secondly, [1] recently showed that PFNs clearly outperform MCMC in terms of runtime and accuracy on the task of learning curve extrapolation, highlighting that generating sufficient samples with MCMC to reliably approximate the posterior may impose significant overhead (see Section 4 in [1]).
>
> We, therefore, decide to base our approach on PFNs rather than MCMC, and to train on synthetic FPT distributions instead of simulation data.
>
> [1] Adriaensen, S., Rakotoarison, H., Müller, S., & Hutter, F. (2024). Efficient bayesian learning curve extrapolation using prior-data fitted networks. Advances in Neural Information Processing Systems, 36.
>
> >Language models as priors can have some pathologies. How does the model behave as the amount of data becomes large? In figure 11 it looks like even with a lot of data the PFN doesn't converge to the true CDF.
>
> We agree with the reviewer that the context can have a substantial impact on the resulting approximations. We tried to account for this by reporting means and standard deviations across 20 different contexts for all the approximations shown in the manuscript. Generally, we observe that the approximation performance is better with more context. In Figure 11 mentioned by the reviewer, the largest context size is 75, which arguably is still relatively small compared to the 1000 (or typically even more) simulations required to obtain reliable CDFs from Kinfold. However, also in Figure 11, we observe a clear trend of improved performance with more context data available, which is also a general pattern shown in Table 6 where we compare KinPFN with different alternative approaches. However, we agree with the reviewer that it might be interesting to see KinPFN’s behavior for much larger context sizes. We, therefore, add a Table with the results for context sizes of up to 1000 to the Appendix H.2 of our revised manuscript.

---

> ### Author Response · Authors · 2024-11-21
> **Response to Reviewer FZT1 continued**
>
> We thank the reviewer for the helpful feedback. If there are any further questions or any clarification needed, we are happy to answer those. If not, we would appreciate it if the reviewer could increase our score.
>
> With kind regards,
>
> The authors

---

> > ### Comment · Reviewer_FZT1 · 2024-11-23
> > **Comment**
> >
> > Ok, it seems using synthetic data is not a good idea.
> >
> > I'm a little confused by the response to the second question. I'm not suggesting running MCMC on the RNA structure, just on the  fit of the multimodal Gaussian data. Fitting these types of models is extremely standard and there is a very large amount of work on them. If they can be applied to the data then it seems they should be cited and compared to PFNs. Could the authors clarify this?
> >
> > For the last question, I would expect Gaussian mixture models to handle large amounts of data just fine.

---

> > > ### Author Response · Authors · 2024-11-25
> > > **Author response to Reviewer FZT1**
> > >
> > > Dear Reviewer FZT1,
> > >
> > > Please see our responses below.
> > >
> > > >Ok, it seems using synthetic data is not a good idea.
> > >
> > > We do not really understand the conclusion drawn by the reviewer and do not agree that training on synthetic data is not a good idea. We would like to kindly ask the reviewer what this conclusion is based on?
> > >
> > > >I'm a little confused by the response to the second question. I'm not suggesting running MCMC on the RNA structure, just on the fit of the multimodal Gaussian data. Fitting these types of models is extremely standard and there is a very large amount of work on them. If they can be applied to the data then it seems they should be cited and compared to PFNs. Could the authors clarify this?
> > >
> > > We are sorry about the reviewer’s confusion due to our response.
> > >
> > > We chose PFNs over MCMC due to the strong performance of PFNs compared to MCMC in the task of learning curve extrapolation shown in [1].
> > >
> > > [1] Adriaensen, S., Rakotoarison, H., Müller, S., & Hutter, F. (2024). Efficient bayesian learning curve extrapolation using prior-data fitted networks. Advances in Neural Information Processing Systems, 36.
> > >
> > > >For the last question, I would expect Gaussian mixture models to handle large amounts of data just fine.
> > >
> > > We add the analysis of the different other methods to Tables 8, 9, and 10 in the revised version of our manuscript. Indeed, it seems like GMMs can handle large contexts quite well, showing the best performance in terms of NLL, however, KinPFN outperforms all  the other methods in terms of MAE and KS also for the large context sizes.
> > >
> > > With kind regards,
> > >
> > > The Authors

---

> > > > ### Comment · Reviewer_FZT1 · 2024-11-25
> > > > **Comment**
> > > >
> > > > Thanks for the response! My statement about the synthetic data was only to say that you've convinced me that my suggestion of training on synthetic data to build a prior was not a sound idea.
> > > >
> > > > WRT the comparison to GMMs I appreciate the authors including tables 7-9. As I understand however, KinPFN was trained to model a GMM prior, so why does it outperform them? Is it because the number of components is variable? If you built a GMM with a prior over the number of components identical to that of KinPFN then how would it perform.

---

> > > > > ### Author Response · Authors · 2024-11-26
> > > > > **Author response**
> > > > >
> > > > > Dear Reviewer FZT1
> > > > >
> > > > > Thanks for the fast response. Please find the details below.
> > > > >
> > > > > >Thanks for the response! My statement about the synthetic data was only to say that you've convinced me that my suggestion of training on synthetic data to build a prior was not a sound idea.
> > > > >
> > > > > We thank the reviewer for the clarification.
> > > > >
> > > > > >WRT the comparison to GMMs I appreciate the authors including tables 7-9. As I understand however, KinPFN was trained to model a GMM prior, so why does it outperform them? Is it because the number of components is variable?
> > > > >
> > > > > Without having investigated this in detail, we would also speculate that the reason is the variable number of components. This is further supported by the better performance of DP-GMMs and KDEs.
> > > > >
> > > > > >If you built a GMM with a prior over the number of components identical to that of KinPFN then how would it perform.
> > > > >
> > > > > KinPFN is trained for 2-5 modes. Therefore, we evaluate GMM baselines with 2-5 components. Similarly, we set the upper bound for DP-GMMs to 2-5 for all comparisons. Our results thus indicate that the performance is worse for both DP-GMM and GMMs when using the same setup.
> > > > >
> > > > > With kind regards,
> > > > >
> > > > > The Authors

---

> > > > > > ### Comment · Reviewer_FZT1 · 2024-11-26
> > > > > > **Comment**
> > > > > >
> > > > > > Thanks for clarifying. What I would like to understand to decide on my score is basically: if it's easy to directly model full Bayesian inference with the prior of interest, why should we pursue approximate inference with a PFN? Your baselines have a fixed number of components but it's not hard to fit 3 GMMs for k=3, 4, 5 and then weight the models according to their marginal likelihoods. What I'm worried about in other words is that your method approximates something we can model directly. This is why I asked about using real data that couldn't be modeled so easily. I would really appreciate your clarifying this!
> > > > > >
> > > > > > The alternative point you made with the Adriaensen paper -- that in some cases the approximation does better than the exact procedure -- is I'm not sure a sound strategy for building a model. For example, if you fit a more flexible model with more compute then you would expect it to model the prior better and therefore do worse.

---

> > > > > > > ### Comment · Reviewer_FZT1 · 2024-11-28
> > > > > > > **Comment**
> > > > > > >
> > > > > > > Thank you for the productive conversation so far! My sense is that the main contribution of this paper is that it suggests fitting a 3 component Gaussian mixture model to a one-dimensional summary statistic from RNA kinetics simulation. While potentially useful and interesting to RNA biologists, I don't believe this represents a significant machine learning contribution. I've adjusted my score to reflect this. On the other hand, despite the abundance of methods to fit these types of models (which also could have been used to tune hyperparameters), the strategy of this paper is to approximate this inference by pre-training an LLM, as suggested by previous works; while this strategy has its pros and cons, it doesn't rise to the level of a fundamental machine learning contribution.

---

> > > > > > > > ### Author Response · Authors · 2024-11-28
> > > > > > > > **Author response**
> > > > > > > >
> > > > > > > > Dear Reviewer FZT1,
> > > > > > > >
> > > > > > > > Again, we would like to thank you for your quick response. We will detail our thoughts on your reply in the following.
> > > > > > > >
> > > > > > > > >Thank you for the productive conversation so far! My sense is that the main contribution of this paper is that it suggests fitting a 3 component Gaussian mixture model to a one-dimensional summary statistic from RNA kinetics simulation. While potentially useful and interesting to RNA biologists, I don't believe this represents a significant machine learning contribution. I've adjusted my score to reflect this. On the other hand, despite the abundance of methods to fit these types of models (which also could have been used to tune hyperparameters), the strategy of this paper is to approximate this inference by pre-training an LLM, as suggested by previous works; while this strategy has its pros and cons, it doesn't rise to the level of a fundamental machine learning contribution.
> > > > > > > >
> > > > > > > > We respectfully disagree with the reviewer's description of our approach. To our knowledge, applications to physical sciences (including chemistry and biology) are explicitly listed as a relevant topic in the ICLR call for papers at https://iclr.cc/Conferences/2025/CallForPapers and our approach describes a significant interdisciplinary contribution to an interesting application from the field of biology. In addition, we do not fit a 3 component Gaussian mixture to the problem of RNA first passage time distribution approximation but develop a new strategy that allows training a prior-data fitted network (PFN) on synthetic first passage time (FPT) distributions without the need for quantile information. Also, the model is not an LLM (with 4.8M parameters not large and no language involved at all). However, the resulting model, KinPFN, is the first deep learning model in the field of RNA folding kinetics analysis. This is a novel contribution not only to the field of PFNs but also to the field of RNA kinetics. However, while we agree that our new prior is arguably simple and describes distributions of parameterized multi-modal Gaussians, our empirical analysis (see also previous response) further indicates that KinPFN learns to approximate OOD data better than GMMs and ensembles of these, resulting in superior performance. Our training on synthetic data for a biological problem further showcases that it is possible to model biological processes even in the absence of real-world examples. We, therefore, think that our new strategy for training and inferring PFNs in the absence of quantile information, the novelty and relevance of the topic, as well as the strong approximation performance which results in massive speed-ups, are reasonable contributions of an application paper at ICLR.
> > > > > > > >
> > > > > > > > In addition, while we would call KinPFN a pioneering work in the field, it shows a large potential to create a substantial impact in an important application area. Since the prior is not limited to a multi-modal Gaussian, there is no need to retrain or refit KinPFN for applications in online learning settings such as kinetic RNA design, and since KinPFN performs stable without further hyperparameters during inference, our method shows several clear advantages over simply fitting a 3 component Gaussian mixture model to the problem.
> > > > > > > >
> > > > > > > > With kind regards,
> > > > > > > >
> > > > > > > > The Authors

---

> > > > > > > > > ### Comment · Reviewer_FZT1 · 2024-11-28
> > > > > > > > > **Comment**
> > > > > > > > >
> > > > > > > > > Thanks again for your quick detailed response!
> > > > > > > > >
> > > > > > > > > Previously you showed evidence that KinPFN outperforms GMMs with fixed k. As I understood, you affirmed my hypothesis that KinPFN outperforms GMMs because it does full Bayesian inference over the number of components. Now you've done more experiments that show more data that KinPFN outperforms even a mixture. I thank you for performing these experiments but here is my issue: KinPFN is trained to approximate a mixture of GMMs, and presumably, if trained long enough with a large enough architecture, would perform identically to them; therefore, if KinPFN outperforms the mixture of GMMs, it is because it has failed to accurately fit the data. This is a little suspect to me as a sound modeling paradigm because 1) it's unclear how to turn this into principles for designing PFN architectures and training these models if you're actively trying to avoid fitting the training data, and 2) it is likely that this mis-fitting will manifest in pathologies in other regimens, say large N.
> > > > > > > > >
> > > > > > > > > Claims that KinPFN could be trained to fit other priors seem to have worked out more poorly that fitting the synthetic data -- you mentioned that the performance was worse when fit to simulation data. For this reason, I don't think this potential flexibility of the PFN should count as an advantage.
> > > > > > > > >
> > > > > > > > > On the other hand, the Adriaensen paper attempted to fit a prior that was much harder to approximate via MCMC; in this case they suggested that a PFN might be a good alternative and showed that in their experiment in their Fig 3 -- they in particular showed that the failure of MCMC was that even with many many samples, the posterior could not be fit. As I understand, this paper makes a different argument: not that MCMC is hard for this data, but that KinPFN generalizes in a useful way when not trained to exactly match; this is a more suspect argument for the reasons outlined above.
> > > > > > > > >
> > > > > > > > > Another point the authors made along the lines of the argument of the Adriaensen paper was that KinPFN is fast while MCMC is time consuming. STAN I believe can fit these models on the order of a minute and there are faster GPU-accelerated libraries as well. For this to could as a substantial contribution, I believe the authors should argue that reducing the fitting time of an RNA kinetics curve down from a minute is useful; given that the kinetics data is not super abundant, what is enabled by being able to analyze these data faster than a minute?
> > > > > > > > >
> > > > > > > > > Also, excuse me for using the term "LLM". I accidentally used it as a metonym for a transformer.
> > > > > > > > >
> > > > > > > > > Finally, I appreciate the importance of interdisciplinary research and I also appreciate the authors working on solving problems in RNA kinetics. However, fitting simple, classical Bayesian models using packages like STAN is a staple of modern computational biology research; these models are regularly fit to microscopy, health, spectroscopy, and all sorts of other data. Therefore, I don't find approximately fitting a 4 component mixture of GMMs to one dimensional data to be a novel machine learning contribution.

---

> > > > > > > > > > ### Author Response · Authors · 2024-11-29
> > > > > > > > > > **Author clarifications**
> > > > > > > > > >
> > > > > > > > > > Dear Reviewer FZT1,
> > > > > > > > > >
> > > > > > > > > > We acknowledge the fruitful discussion and thank you again for the response.
> > > > > > > > > >
> > > > > > > > > > Again, please see below for our detailed responses.
> > > > > > > > > >
> > > > > > > > > > >Previously you showed evidence that KinPFN outperforms GMMs with fixed k. As I understood, you affirmed my hypothesis that
> > > > > > > > > > KinPFN outperforms GMMs because it does full Bayesian inference over the number of components.
> > > > > > > > > >
> > > > > > > > > > We agree that ‘standard’ GMMs might perform badly on the data due to their limitation to a fixed component.

---

> > > > > > > > > > ### Author Response · Authors · 2024-11-29
> > > > > > > > > > **Author clarifications continued**
> > > > > > > > > >
> > > > > > > > > > >Now you've done more experiments that show more data that KinPFN outperforms even a mixture. I thank you for performing these experiments but here is my issue: KinPFN is trained to approximate a mixture of GMMs, and presumably, if trained long enough with a large enough architecture, would perform identically to them; therefore, if KinPFN outperforms the mixture of GMMs, it is because it has failed to accurately fit the data. This is a little suspect to me as a sound modeling paradigm because 1) it's unclear how to turn this into principles for designing PFN architectures and training these models if you're actively trying to avoid fitting the training data, and 2) it is likely that this mis-fitting will manifest in pathologies in other regimens, say large N.
> > > > > > > > > >
> > > > > > > > > > We think the Reviewer’s conclusion that KinPFN failed to accurately fit the prior and therefore achieves good performance on real simulation data is not the only interpretation of our results and further does **not** align very well with our approach nor our empirical results. We would like to challenge this view with a different interpretation that aligns much better with our setup, recent observations, and our empirical results by addressing the following three questions:
> > > > > > > > > >
> > > > > > > > > > 1. Does KinPFN accurately fit the prior?
> > > > > > > > > > 2. What can cause the worse performance of the GMM ensemble on samples drawn from the prior?
> > > > > > > > > > 3. What explains the strong performance of KinPFN on the real simulation data?
> > > > > > > > > >
> > > > > > > > > > ### 1. Does KinPFN accurately fit the prior?
> > > > > > > > > >
> > > > > > > > > > We strongly disagree with the reviewer that KinPFN is **not** trained to fit the prior but to generalize better to new distributions. While we have to admit that HPO was performed on the newly introduced validation set of real simulation data, we did not use the validation performance for early stopping our training. Rather, we train KinPFN to ‘convergence’, or better, until the NLL varies only slightly within a given epsilon due to the infinite nature of the data. This also means that we have learned representations of the contexts whose combination results in minimizing the KL between the model prediction and the prior sufficiently since we are using NLL in the infinite data regimen (for a proof see [1]; https://openreview.net/forum?id=KSugKcbNf9 ). Summarizing, we fit the prior well before continuing with further experiments.
> > > > > > > > > >
> > > > > > > > > > This is also evidenced by the strong empirical results of KinPFN on the examples drawn from the prior and we account for the influence of different contexts by reporting the mean and the standard-deviation across 20 different context inputs for all approximations shown in the manuscript.
> > > > > > > > > >
> > > > > > > > > > That being said, we were wondering if we aim at training KinPFN to not accurately fit the prior, wouldn’t that result in worse performance on samples drawn from the prior compared to a GMM ensemble? This is not the case and thus, we think that the interpretation of the reviewer is not well aligned with our empirical results.
> > > > > > > > > >
> > > > > > > > > > However, we cannot assume that KL is zero for all inputs because we can only observe a small fraction of all data even when training forever using a larger model. Thus, even if the prior is a mixture of GMMs, we cannot assume that endless training on it would result in exact predictions that match the predictions of a GMM ensemble.
> > > > > > > > > >
> > > > > > > > > > This leads to the question why the GMM ensemble is performing worse than KinPFN on samples drawn from the prior?
> > > > > > > > > >
> > > > > > > > > > ### 2. What can cause the worse performance of the GMM ensemble on samples drawn from the prior?
> > > > > > > > > >
> > > > > > > > > > While we agree that we implement a parameterized multi-modal Gaussian prior distribution, it is possible that the prior might not always be optimal for approximations with a GMM since we did not carefully construct the prior to work well with GMMs but to learn reasonable representations in the PFN. For instance, the synthetic prior may involve overlapping or poorly separated components, making it challenging for GMMs to assign weights and means correctly. In contrast, KinPFN does not directly rely on GMM components or explicit parameterization of the prior during inference.
> > > > > > > > > >
> > > > > > > > > > In addition, while the ensemble approach improves flexibility by using multiple component GMMs and weighting them by marginal likelihoods, it still requires explicit marginalization and proper model selection which might not be optimal and could require tuning.

---

> > > > > > > > > > ### Author Response · Authors · 2024-11-29
> > > > > > > > > > **Author clarifications continued**
> > > > > > > > > >
> > > > > > > > > > ### 3. What explains the strong performance of KinPFN on the real simulation data?
> > > > > > > > > >
> > > > > > > > > > While we are aware of work showing that ICL is a transient phenomena when training a transformer and that it competes with in-weight learning (IWL) (see e.g. [2]; https://arxiv.org/pdf/2311.08360), recent results show that this is not necessarily the case in the PFN setting.
> > > > > > > > > >
> > > > > > > > > > For example, it was recently shown that a PFN trained only on step-functions can generalize to smooth predictions while fitting the prior well (see https://x.com/SamuelMullr/status/1841907948219727984 ; on X, yes, but still a nice result).
> > > > > > > > > >
> > > > > > > > > > Given these insights, we can assume that KinPFN can generalize to distributions different from the prior, as indicated empirically with strong performance on the real simulation data, outperforming GMMs and ensembles of these.
> > > > > > > > > >
> > > > > > > > > > However, to further support this claim, we construct a simple synthetic example, using a multi-modal uniform distribution of the prior instead of the multi-modal Gaussian. We evaluate the ensemble of GMMs and KinPFN on 10,000 samples drawn from this (similar but clearly different) prior. The results are shown below.
> > > > > > > > > >
> > > > > > > > > > | Context Size | Model | MAE | NLL |
> > > > > > > > > > |----------------|-----------------------------|-----------------------------|----------------|
> > > > > > > > > > | 10      | KinPFN            |   **0.086**               |         **1.462**                                       |
> > > > > > > > > > |           | GMM Ensemble | 0.118   	| 7.085 |
> > > > > > > > > > | 25      | KinPFN            |   **0.062**               |    **1.314** |
> > > > > > > > > > |          | GMM Ensemble  | 0.106   	| 1.685               	|
> > > > > > > > > > | 50      | KinPFN            |   **0.051**               |     1.281                                           |
> > > > > > > > > > |           | GMM Ensemble       	| 0.102    	| **1.033**      	|
> > > > > > > > > > | 75      | KinPFN            |   **0.046**               |  1.271                                              |
> > > > > > > > > > |          |     GMM Ensemble   	| 0.099   	| **0.935**       	|
> > > > > > > > > > | 100 | KinPFN       	| **0.044**   	| 1.266       	|
> > > > > > > > > > |     | GMM Ensemble       	| 0.098   	| **0.901**       	|
> > > > > > > > > >
> > > > > > > > > > These results support that KinPFN can generalize to a different distribution than defined in the prior.
> > > > > > > > > >
> > > > > > > > > > However, we agree with the Reviewer that a clear requirement to achieve this generalization is that the prior supports reasonable approximations of the posterior. In other words, if we cannot learn a good representation of the context that would also provide good approximations when using context from the posterior, the learned representations get exponentially worse during training (due to the multiplicative form of the likelihood). Our model would get more confident in a wrong representation, which would lead to worse performance when adding more and more context (larger N). However, this is not the case for KinPFN as we show empirically in Tables 8-10 where the performance of KinPFN constantly improves even with very large N.
> > > > > > > > > >
> > > > > > > > > > We thus conclude that (1) KinPFN has learned good representations from the prior that support approximations of the posterior and (2) KinPFN can generalize to different distributions outside of the prior.
> > > > > > > > > >
> > > > > > > > > > In summary, we think that the interpretation that KinPFN has learned good representations of the prior that enable generalization to the similar but different real simulation posterior, aligns much better with our empirical findings than the interpretation of the Reviewer that strong performance of KinPFN is a result of inaccurate fitting of the prior during training.
> > > > > > > > > >
> > > > > > > > > > This could even be formulated as a contribution since our results allow us to derive principles that help training PFNs for a given task: We provide new evidence that PFNs and ICL can lead to generalization to new distributions, while also showing the necessity to carefully check the performance on similar but different distributions, particularly for larger Ns.

---

> > > > > > > > > > ### Author Response · Authors · 2024-11-29
> > > > > > > > > > **Author clarifications continued**
> > > > > > > > > >
> > > > > > > > > > >Claims that KinPFN could be trained to fit other priors seem to have worked out more poorly that fitting the synthetic data -- you mentioned that the performance was worse when fit to simulation data. For this reason, I don't think this potential flexibility of the PFN should count as an advantage.
> > > > > > > > > >
> > > > > > > > > > As mentioned before, we tried training KinPFN on simulation data for relatively short sequences (up to 30 nucleotides) since simulations for longer sequences are infeasible due to long runtimes. As also mentioned in the same response, the conformation space grows exponentially with the sequence length and we, therefore, expect that the real-world prior does not fit the true posterior very well. Furthermore, we are moving away from the infinite data regime back to a setting where the training results are bound by data availability rather than being compute bound as in the infinite data setting. We thus have to avoid overfitting and might have to use strong regularization to achieve good results. We did not further follow up on this approach because the data is not available at scale. However, we do not think that our bad preliminary results on real simulation data are related to not fitting the distribution but rather they are mainly due to limited data availability.
> > > > > > > > > >
> > > > > > > > > > We, therefore, still think that a synthetic prior that fits the true posterior better than the current one, e.g. by using other distributions than multi-modal Gaussians, could still lead to performance improvements. However, there has to be support for good approximations of the posterior in the prior as mentioned before.
> > > > > > > > > >
> > > > > > > > > > >On the other hand, the Adriaensen paper attempted to fit a prior that was much harder to approximate via MCMC; in this case they suggested that a PFN might be a good alternative and showed that in their experiment in their Fig 3 -- they in particular showed that the failure of MCMC was that even with many many samples, the posterior could not be fit. As I understand, this paper makes a different argument: not that MCMC is hard for this data, but that KinPFN generalizes in a useful way when not trained to exactly match; this is a more suspect argument for the reasons outlined above.
> > > > > > > > > >
> > > > > > > > > > As outlined in detail above, we do not agree that KinPFN is trained to not fit the prior. However, we agree that we think that KinPFN can generalize to new distributions.
> > > > > > > > > >
> > > > > > > > > > >Another point the authors made along the lines of the argument of the Adriaensen paper was that KinPFN is fast while MCMC is time consuming. STAN I believe can fit these models on the order of a minute and there are faster GPU-accelerated libraries as well. For this to could as a substantial contribution, I believe the authors should argue that reducing the fitting time of an RNA kinetics curve down from a minute is useful; given that the kinetics data is not super abundant, what is enabled by being able to analyze these data faster than a minute?
> > > > > > > > > >
> > > > > > > > > > We agree with the Reviewer that this might not be a major advantage of KinPFN over these methods in our case. However, we also do not see a good reason to use an alternative approach to PFNs when we can assume that the methods will be orders of magnitudes slower (as indicated by the runtime of 1 minute mentioned by the reviewer compared to a single forward pass) before having run a single experiment. We, therefore, think it is still valid to prioritize a PFN approach over other solutions (which also have never been used to speed-up RNA kinetics simulations before!) for the development of a method for an application where there is no previous knowledge available.

---

> > > > > > > > > > ### Author Response · Authors · 2024-11-29
> > > > > > > > > > **Author clarifications continued**
> > > > > > > > > >
> > > > > > > > > > >Also, excuse me for using the term "LLM". I accidentally used it as a metonym for a transformer.
> > > > > > > > > >
> > > > > > > > > > We thank the reviewer for the excuse. We mentioned this minor issue as we think the term LLM is quite biased these days and might lead to misunderstandings.
> > > > > > > > > >
> > > > > > > > > > >Finally, I appreciate the importance of interdisciplinary research and I also appreciate the authors working on solving problems in RNA kinetics. However, fitting simple, classical Bayesian models using packages like STAN is a staple of modern computational biology research; these models are regularly fit to microscopy, health, spectroscopy, and all sorts of other data. Therefore, I don't find approximately fitting a 4 component mixture of GMMs to one dimensional data to be a novel machine learning contribution.
> > > > > > > > > >
> > > > > > > > > > Since KinPFN is the first model in the field, we would be very happy if future approaches challenge our results with different approaches. However, we think that using a PFN for our problem is also a valid approach for the reasons described above.
> > > > > > > > > >
> > > > > > > > > > [1] Müller, S., Hollmann, N., Arango, S. P., Grabocka, J., & Hutter, F. Transformers Can Do Bayesian Inference. In International Conference on Learning Representations.
> > > > > > > > > >
> > > > > > > > > > [2] Singh, A., Chan, S., Moskovitz, T., Grant, E., Saxe, A., & Hill, F. (2024). The transient nature of emergent in-context learning in transformers. Advances in Neural Information Processing Systems, 36.
> > > > > > > > > >
> > > > > > > > > > We thank the reviewer for the helpful and interesting discussion that helps us improve our manuscript. We hope that our explanations help to clarify the Reviewer’s concerns and questions. However, we are happy to further discuss the advantages of ICL with PFNs for the problem at hand if necessary. If this is not the case, we would like to kindly ask the Reviewer to consider updating our score.
> > > > > > > > > >
> > > > > > > > > > With kind regards,
> > > > > > > > > >
> > > > > > > > > > The Authors

---

> > > > > > > > > > > ### Comment · Reviewer_FZT1 · 2024-11-29
> > > > > > > > > > > **Comment**
> > > > > > > > > > >
> > > > > > > > > > > I’m not sure the authors and I are in the same page. I’m claiming: kinPFN is trained to approximate a GMM prior. Therefore, if KinPFN performs this approximation well, then it will behave exactly as the GMM prior when simulated with MCMC. I’m unsure if you’re suggesting that somehow the model will generalize in a different way when the prior is misspecified because of the properties of ICL, but I don’t think this is likely: KinPFN should behave just like a GMM would if your conviction of it’s being a good approximation is true.

---

> > > > > > > > > > > > ### Author Response · Authors · 2024-11-30
> > > > > > > > > > > > **Author response**
> > > > > > > > > > > >
> > > > > > > > > > > > Dear Reviewer FZT1,
> > > > > > > > > > > >
> > > > > > > > > > > > We apologize that our previous conversation led to confusion.
> > > > > > > > > > > >
> > > > > > > > > > > > We train KinPFN on roughly 5 million GMMs. For each of these GMMs we know the exact parametrization and these GMMs would exactly approximate the prior distributions. However, at test time, when fitting a GMM on the provided context, this GMM doesn’t have to be optimal. There are failure cases when fitting GMMs, for example when the distribution of interest contains overlapping modes. This becomes worse when using an ensemble of different GMMs, since the ensemble comes with its own limitations.
> > > > > > > > > > > >
> > > > > > > > > > > > It is likely that the training examples contain cases where fitting a GMM would lead to suboptimal results (e.g. in the case of poorly separated modes). The PFN might thus handle these cases better due to the massive amounts of different GMMs it was trained on and the benefits of the learned representations.
> > > > > > > > > > > >
> > > > > > > > > > > > Therefore, we disagree that KinPFN has to behave exactly like a GMM fitted on the context but can likely perform better as indicated by our results.
> > > > > > > > > > > >
> > > > > > > > > > > > With kind regards,
> > > > > > > > > > > >
> > > > > > > > > > > > The Authors

---

> > > > > > > > > > > > > ### Comment · Reviewer_FZT1 · 2024-11-30
> > > > > > > > > > > > > **Comment**
> > > > > > > > > > > > >
> > > > > > > > > > > > > KinPFN is indeed trained on 5 million GMM, which each have their parameters drawn from a prior. This describes a prior over GMMs and training the PFN to minimize this likelihood is guaranteed to make it approximate the full Bayesian model. When given new data, a Bayesian GMM will infer all the parameters, such as the means and stds and. number of components using Bayes rule; this can be approximated by MCMC quite easily (ex https://arxiv.org/abs/1502.06241).
> > > > > > > > > > > > >
> > > > > > > > > > > > > The Bayes-optimal model, the one that maximizes likelihood on the data you are training KinPFN on, is this Bayesian model. Therefore, when you train KinPFN, it approaches this model and in particular should behave just like it. If KinPFN outperforms this model then it must be that it doesn't optimize its training objective.
> > > > > > > > > > > > >
> > > > > > > > > > > > > I appreciate the author's extended discussion but I don't think there is any use in continuing this discussion.

---

> > > > > > > > > > > > > > ### Author Response · Authors · 2024-12-01
> > > > > > > > > > > > > > **Clarifications from senior author**
> > > > > > > > > > > > > >
> > > > > > > > > > > > > > Dear Reviewer FZT1,
> > > > > > > > > > > > > >
> > > > > > > > > > > > > > Senior author here. Thank you for your engagement! However, I believe we failed to clarify some fundamental points with you:
> > > > > > > > > > > > > >
> > > > > > > > > > > > > > - Both MCMC and PFNs approximate the exact Bayesian posterior. They both do full Bayesian inference. It’s just that PFNs are dramatically faster and often yield a much better approximation than MCMC with reasonable time limits.
> > > > > > > > > > > > > >
> > > > > > > > > > > > > > - Two previous papers have directly compared PFNs and MCMC and shown PFNs to be 10000x faster, or much better at the same computational cost: 1. https://arxiv.org/abs/2112.10510 (the paper introducing PFNs; see Figure 5 for 10000x faster convergence to the same performance) and 2. https://proceedings.neurips.cc/paper_files/paper/2023/file/3f1a5e8bfcc3005724d246abe454c1e5-Paper-Conference.pdf (the LC-PFN paper we mentioned earlier in this rebuttal; see Figure 10 for convergence to a solution in 0.1s that MCMC does not reach in 1000s but would eventually reach with optimal hyperparameters and enough time).
> > > > > > > > > > > > > >
> > > > > > > > > > > > > > - While MCMC has a rich theory and history, it is also conceptually complex (our problem with a varying number of mixture components could, e.g., not be addressed by standard MCMC, but would require extensions like reversible jump MCMC), is nontrivial to get right (choosing various hyperparameters, proposal distributions, burn-in, etc) and computationally very costly.
> > > > > > > > > > > > > >
> > > > > > > > > > > > > > - For the full background on PFNs, see https://arxiv.org/abs/2112.10510. Insight 1, Corollary 1.1 and Corollary 1.2 there show theoretically that by optimizing cross entropy loss PFNs directly minimize approximation error of the posterior. Perfectly optimizing that cross entropy loss (hypothetically: with infinite data, infinite compute and the right optimizer) leads to the approximation of the posterior being exact. Figure 3 (right) in the same paper demonstrates very truthful approximations empirically with finite data and time, and actual optimizers (on a Gaussian process, where the exact posterior is available in closed form).
> > > > > > > > > > > > > >
> > > > > > > > > > > > > > - So far, no MCMC solution exists for the problem of predicting RNA folding time distributions we’re tackling. We agree that it could also be sensible to use MCMC, but we chose PFNs since it’s a much better fit for the problem and much easier to do (no need for reversible jump MCMC, etc). We strongly believe that we should not be penalized for not comparing to a method that hasn’t been used for this problem before.
> > > > > > > > > > > > > >
> > > > > > > > > > > > > > - For completeness and the avoidance of doubt, we note that PFNs also have key disadvantages compared to MCMC, in particular not giving access to the samples from the latent. But for cases where these are not needed (like the current) they are often the best choice.
> > > > > > > > > > > > > >
> > > > > > > > > > > > > > We hope to have clarified these points and would be glad to address any follow-up questions.

---

> > > > > > > ### Author Response · Authors · 2024-11-28
> > > > > > > **Author clarifications**
> > > > > > >
> > > > > > > Dear Reviewer FZT1,
> > > > > > >
> > > > > > > We thank you for your fast responses. Please see our clarifications below.
> > > > > > >
> > > > > > > >Thanks for clarifying. What I would like to understand to decide on my score is basically: if it's easy to directly model full Bayesian inference with the prior of interest, why should we pursue approximate inference with a PFN? Your baselines have a fixed number of components but it's not hard to fit 3 GMMs for k=3, 4, 5 and then weight the models according to their marginal likelihoods. What I'm worried about in other words is that your method approximates something we can model directly. This is why I asked about using real data that couldn't be modeled so easily. I would really appreciate your clarifying this!
> > > > > > >
> > > > > > > We thank the reviewer for this interesting and important question.
> > > > > > >
> > > > > > > Since we cannot assume that the distribution of first passage times can be fully represented as a multi-modal Gaussian, it is likely that the GMM cannot infer it exactly. However, we agree that an ensemble of GMMs for different components (weighted according to their marginal likelihoods) could improve flexibility and performance. We thus implement it and compare KinPFN to the GMM ensemble on the testset of 635 randomly generated sequences with KinFold simulations. The results are shown below.
> > > > > > >
> > > > > > > Ensemble for components k = 2,3,4,5:
> > > > > > >
> > > > > > > | Context Size | Model | MAE | NLL |
> > > > > > > | -------------- | -------- | ------ | ------- |
> > > > > > > | 10 | KinPFN | **0.084** | **1.374** |
> > > > > > > |       | GMM Ensemble | 0.093 | 6.417 |
> > > > > > > | 25 | KinPFN | **0.056** | **1.244** |
> > > > > > > |       | GMM Ensemble | 0.084 | 1.917 |
> > > > > > > | 50 | KinPFN | **0.039** | **1.205** |
> > > > > > > |      |  GMM Ensemble | 0.081 | 1.389 |
> > > > > > > | 75  | KinPFN | **0.033** | **1.192** |
> > > > > > > |       | GMM Ensemble | 0.080 | 1.261 |
> > > > > > > | 100 | KinPFN | **0.030** | **1.186** |
> > > > > > > |      | GMM Ensemble | 0.078 | 1.218 |
> > > > > > >
> > > > > > > For components k = 2,3,4:
> > > > > > >
> > > > > > > | Context Size | Model | MAE | NLL |
> > > > > > > | ------------- | -------- | ------ | ------ |
> > > > > > > | 10 | KinPFN | **0.084** | **1.374** |
> > > > > > > |       | GMM Ensemble | 0.095 | 4.270 |
> > > > > > > | 25 | KinPFN | **0.056** | **1.244** |
> > > > > > > |       | GMM Ensemble | 0.086 | 1.639 |
> > > > > > > | 50 | KinPFN | **0.039** | **1.205** |
> > > > > > > |      |  GMM Ensemble | 0.081 | 1.312 |
> > > > > > > | 75  | KinPFN | **0.0333** | **1.192** |
> > > > > > > |       | GMM Ensemble | 0.080 | 1.228 |
> > > > > > > | 100 | KinPFN | **0.030** | **1.186** |
> > > > > > > |      | GMM Ensemble | 0.078 | 1.202 |
> > > > > > >
> > > > > > > KinPFN clearly outperforms both ensemble variants across all context sizes.
> > > > > > >
> > > > > > > We also evaluate a GMM ensemble (k = 2,3,4,5 matching the KinPFN training modes) on 10,000 samples directly from the prior:
> > > > > > >
> > > > > > > | Context Size | Model | MAE | NLL |
> > > > > > > |----------------|-----------------------------|-----------------------------|----------------|
> > > > > > > | 10      | KinPFN            |  **0.088**                |        **2.427**                                        |
> > > > > > > |           | GMM Ensemble | 0.103    	| 7.386 |
> > > > > > > | 25      | KinPFN            |  **0.055**                |        **2.136**                                        |
> > > > > > > |          | GMM Ensemble  | 0.0714   	| 2.732               	|
> > > > > > > | 50      | KinPFN            |              **0.039**    |                   **2.060**                             |
> > > > > > > |           | GMM Ensemble       	| 0.062    	| 2.204      	|
> > > > > > > | 75      | KinPFN            |         **0.032**         |    **2.039**                                            |
> > > > > > > |          |     GMM Ensemble   	| 0.059   	| 2.100       	|
> > > > > > > | 100    | KinPFN     | **0.028**  | **2.028**  |
> > > > > > > |          | GMM Ensemble        	| 0.057   	| 2.061       	|
> > > > > > >
> > > > > > > We think that these results allow for two conclusions: (1) It doesn’t seem to be the case that an ensemble of GMMs can approximate the data similarly well as KinPFN and (2) KinPFN does not only mimic the behavior of GMMs. The reason might be that KinPFN does not directly rely on GMM components or explicit parameterization of the prior during inference. Instead, it is trained on a distribution of multi-modal priors and might learn to generalize beyond individual instances of these priors. This could allow KinPFN to better capture subtle features of the synthetic prior, such as complex dependencies between modes or variations in the structure of the modes during training. In essence, KinPFN learns a more effective representation of the prior during training and therefore generalizes better to the real-world data.
> > > > > > >
> > > > > > > That being said, we would like to emphasize some more advantages of KinPFN over GMMs:
> > > > > > >
> > > > > > > - In contrast to GMMs, KinPFN is not limited to multi-modal Gaussian distributions but can be trained on different prior distributions or even mixtures of them.
> > > > > > > - There is no need to retrain or refit KinPFN and there is only a single forward pass required to approximate a given distribution. This is particularly important for applications such as kinetic RNA design, where KinPFN could be used on top of an oracle (e.g. KinFold).
> > > > > > > - KinPFN performs stable and does not require hyperparameters at test time.
> > > > > > >
> > > > > > > We hope this clarifies the reviewer’s question.

---

> > > > > > > ### Author Response · Authors · 2024-11-28
> > > > > > > **Author clarifications continued**
> > > > > > >
> > > > > > > >The alternative point you made with the Adriaensen paper -- that in some cases the approximation does better than the exact procedure -- is I'm not sure a sound strategy for building a model. For example, if you fit a more flexible model with more compute then you would expect it to model the prior better and therefore do worse.
> > > > > > >
> > > > > > > We think that this mainly depends on the chosen prior. PFNs perform strongly in scenarios where the posterior can be effectively represented by the prior distribution, even when the prior is not an exact match for the data. The strength of PFNs lies in their ability to generalize across distributions during training, leveraging their flexibility to approximate the posterior well without overfitting to specific prior instances.
> > > > > > >
> > > > > > > While we acknowledge that our current multi-modal Gaussian prior may not perfectly represent the true posterior of FPTs, it serves as a reasonable approximation for evaluating PFN performance. We think that this is also substantiated by our empirical results, where KinPFN outperforms the other approaches.
> > > > > > >
> > > > > > > We hope we addressed all the questions of the reviewer. We thank you again for the useful comments and questions that helped us improve our manuscript. If you have any further questions, we are happy to answer them.
> > > > > > >
> > > > > > > With kind regards,
> > > > > > >
> > > > > > > The Authors

---

### Official Review · Reviewer_DNAm · 2024-11-04

**Soundness:** 3
**Presentation:** 2
**Contribution:** 3
**Rating:** 6
**Confidence:** 3

**Summary:**

This paper studies the RNA folding kinetics modeling problem, which is helpful for understanding RNA behavior and designing RNA. The authors propose to apply a deep learning method based on a prior-data fitted network to quickly estimate the distribution of RNA folding times. Experiments in synthetic datasets and real examples demonstrate the effectiveness of the proposed method.

**Strengths:**

1.	This paper is the first to use deep learning methods to study RNA folding kinetics modeling, an important problem in RNA biology.
2.	On synthetic datasets, the proposed method demonstrates superior performance compared to traditional approaches such as kernel density estimation and Gaussian mixture models.
3.	The proposed method has an advantage in running speed.
4.	The paper provides comprehensive details on dataset construction and model training, ensuring high reproducibility.

**Weaknesses:**

1.	The current writing does not facilitate quick comprehension of the research problem for readers from diverse backgrounds. It would be beneficial if the author could include a figure illustrating specific data and formalization when introducing the problem. For instance, depicting the relationship between the RNA folding process and the corresponding change in folding fraction could enhance clarity.
2.	The unique challenges RNA folding kinetics pose are not adequately summarized in the introduction. Additionally, the paper directly employs prior-data fitted networks to model the CDF without additional enhancements. Highlighting the improvements made to address the specific issues in this field would enhance the paper's contribution.
3.	It would be clearer to explicitly state in the introduction or background section whether the paper focuses on RNA's tertiary or secondary structure, and how the folding ratio is calculated.
4.	Compared to the dynamic changes in RNA secondary or tertiary structures, the folding ratio provides very coarse-grained information about RNA folding dynamics, which seems still far from practical applications. The paper needs to further elucidate how this study can contribute to solving RNA biology problems.
5.	In the experimental section, the results from Kinfold are used for validation, but the inherent error of Kinfold needs rigorous demonstration, which diminishes the persuasiveness of the results. Is it possible to use collected or published wet lab data for the evaluation of this problem?
6.	tRNA and rRNA are the most common and numerous types of RNA. It would be better to test a broader and more diverse range of RNA types.
7.	There is a lack of research and discussion on deep learning methods suitable for the data in this problem. The paper only presents the prior-data fitted network for deep learning-based probability density estimation.

**Questions:**

Please refer to the Weaknesses section for details.

---

> ### Author Response · Authors · 2024-11-21
> **Response to Reviewer DNAm**
>
> Dear Reviewer DNAm,
>
> Thank you for your valuable feedback and for highlighting the novelty and reproducibility of our approach. In the following, we address your concerns and questions in detail.
>
> >The current writing does not facilitate quick comprehension of the research problem for readers from diverse backgrounds. It would be beneficial if the author could include a figure illustrating specific data and formalization when introducing the problem. For instance, depicting the relationship between the RNA folding process and the corresponding change in folding fraction could enhance clarity.
>
> We thank the reviewer for this helpful comment. We are happy to include an overview figure to increase the clarity of our proposed approach. We add a draft for the figure in the Introduction of our revised manuscript.
>
> Would a full version of this figure solve the reviewers’ concerns regarding quick comprehension of the research problem?
>
> >The unique challenges RNA folding kinetics pose are not adequately summarized in the introduction. Additionally, the paper directly employs prior-data fitted networks to model the CDF without additional enhancements. Highlighting the improvements made to address the specific issues in this field would enhance the paper's contribution.
>
> We thank the reviewer for the useful comment. We update the Introduction to clarify our contributions more and to avoid confusion. However, we disagree with the reviewer that we directly employ PFNs to model the CDF without additional enhancements. In contrast to previous work that use PFNs e.g. to extrapolate learning curves, we do not predict the PPD of a target y for a given quantile x conditional on a dataset D, but learn the entire PPD of y (in this case, the first passage times) without knowledge about the quantiles (x is always a zero-vector), effectively representing the absence of further information.
>
> This approach is motivated by the underlying problem structure, where we do not have access to the true quantiles of the context first-passage times. Therefore, we cannot treat the task as a standard regression problem but directly learn the PPD of first passage times conditional on a (data)set of context first passage times but without requiring quantile information which is novel in the field of PFNs.
>
> We add a small part at the end of the Background section of the revised manuscript to point out this novelty more clearly.
>
> >It would be clearer to explicitly state in the introduction or background section whether the paper focuses on RNA's tertiary or secondary structure, and how the folding ratio is calculated.
>
> We include the clarification that the first passage times discussed in the paper are consistently derived from folding simulations that specifically focus on secondary structure formation in the Background section of our revised manuscript.
>
> However, we would like to note that the underlying structural information, be it tertiary or secondary structure based, is more related to the simulators used. As a pure in-context learning approach, KinPFN is well-suited to generalize across different simulators and we would expect that the usage of a tertiary-structure-based simulator for the generation of context FPTs would only marginally impact the FPT approximations of KinPFN.
>
> To clarify the calculation of the folding fraction: the folding fraction is represented by the cumulative distribution function (CDF) derived from a given set of FPTs. To compute the CDF using an available dataset of FPTs—referred to in the paper as the ground truth CDF over a statistically reasonable number of 1000 available FPTs—the process is as follows: The RNA folding times are first arranged in ascending order. For each time point in this sorted list, the fraction of folding events completed by that time is determined by dividing the number of folding times less than or equal to that time by the total number of events. This yields cumulative proportions at each time point, providing a stepwise function that describes the progression of completed folding events over time.

---

> ### Author Response · Authors · 2024-11-21
> **Response to Reviewer DNAm continued**
>
> >Compared to the dynamic changes in RNA secondary or tertiary structures, the folding ratio provides very coarse-grained information about RNA folding dynamics, which seems still far from practical applications. The paper needs to further elucidate how this study can contribute to solving RNA biology problems.
>
> We agree with the reviewer that RNA folding kinetics and dynamics can be captured at different levels of granularity. However, we disagree with the reviewer that first passage times are far from practical applications. In our initial submission, we already show an interesting application, assessing the folding efficiency of different RNA sequences that fold into a common minimum free energy structure (Section 5.3). This kind of analysis is particularly useful for RNA drug discovery, where the rates of obtaining the functional molecular folds could be essential to rank different candidates. In addition, the first passage times of systems play a substantial role in biology, chemistry, and medicine (see e.g. [1]). While KinPFN was primarily developed to study RNA folding kinetics, our results for gene expression data suggest that KinPFN might generalize to other data sources of FPTs as well, with the potential to impact different areas of biology besides RNA folding kinetics.
>
> In Addition, many published examples show the direct application of kinetic folding simulations to biological data, e.g. the original Kinfold paper (doi: 10.1017/s1355838200992161) or studies that investigate riboswitch folding (doi:10.1021/jacs.6b10429)
>
> That said, we add a sentence to the Introduction of the revised version of our manuscript to clearly highlight fields of applications of KinPFN.
>
> [1] Polizzi, N. F., Therien, M. J., & Beratan, D. N. (2016). Mean first‐passage times in biology. Israel journal of chemistry, 56(9-10), 816-824.
>
> >In the experimental section, the results from Kinfold are used for validation, but the inherent error of Kinfold needs rigorous demonstration, which diminishes the persuasiveness of the results. Is it possible to use collected or published wet lab data for the evaluation of this problem?
>
> We agree with the reviewer that we use simulation data obtained from Kinfold as ground truth. However, we also analyze the performance of KinPFN for simulation data from another simulator (Kfold, see experiment 5.1), showing no decrease in the performance of KinPFN. Since KinPFN was not trained using simulation data but a synthetic prior, the performance of KinPFN is independent of the underlying data-generating process. That said, we would expect that KinPFN is capable of predicting first-passage time distributions for wet-lab kinetics data as well. This is also supported by our experiments for gene expression data which arguably requires more transfer capabilities than experimentally obtained first passage times compared to simulation data. However, we are not aware of any resource for obtaining RNA folding kinetics wet-lab data.
>
> >tRNA and rRNA are the most common and numerous types of RNA. It would be better to test a broader and more diverse range of RNA types.
>
> While we agree with the reviewer that tRNA and rRNA are common and well studied RNAs, this was exactly our motivation to use these two types of well-known, structured non-coding RNAs as a reference for KinPFN’s behavior on eukaryotic RNAs. However, in response to the reviewer’s concerns, we are currently running simulations for the following three RNAs:
>
> - https://rnacentral.org/rna/URS00002F3927/224308
> - https://rnacentral.org/rna/URS0000BA5588/9606
> - https://rnacentral.org/rna/URS0000759FB2/9606
>
> We hope that we can obtain the required number of simulations within the next few days.
>
> Would an additional evaluation on these samples resolve the reviewers concerns regarding evaluations for different RNA types?
>
> >There is a lack of research and discussion on deep learning methods suitable for the data in this problem. The paper only presents the prior-data fitted network for deep learning-based probability density estimation.
>
> We thank the reviewer for the useful comment. We add a section on suitable deep learning methods to the related work sections in the main paper and the appendix and further briefly discuss the usage of AI-based methods for MD simulations.
>
> We thank the reviewer again for the valuable feedback and helpful comments. We hope that our response clarified the questions and solved the reviewers’ concerns. If there are any further questions or clarifications required, we are happy to answer those. Otherwise, we would appreciate it if the reviewer could increase our score.
>
> With kind regards,
>
> The authors

---

> > ### Comment · Reviewer_DNAm · 2024-11-21
> >
> > I acknowledge the efforts made to address concerns raised in my initial review. The clarifications provided have improved my understanding of the work. I have decided to revise my score to reflect the progress made.

---

> > > ### Author Response · Authors · 2024-11-21
> > > **Author response to Reviewer DNAm**
> > >
> > > We thank the reviewer for acknowledging our efforts and for increasing our score. If there are any further questions, we are happy to answer them.
> > >
> > > With kind regards,
> > >
> > > The authors

---

### Official Review · Reviewer_PRsD · 2024-11-04

**Soundness:** 3
**Presentation:** 3
**Contribution:** 3
**Rating:** 8
**Confidence:** 3

**Summary:**

The authors propose a novel method to approximate RNA first passage times (FPTs) using Prior-data fitted networks (PFNs), which are transformers that return a posterior-predictive distribution subject to simulated draws from a prior dataset. In this case, the prior dastset is created from a synthetic prior. The authors put a great deal of effort into hand-crafting this prior (e.g. using biological prior knowledge) that will then be used to train their PFN. The result is a prior that is able to train a PFN that predicts FPT better than competing methods, such as KDE, GMM, and DP-GMMs. The paper's biological motivation is sound and the achievement would help researchers in this field. However, the paper has room for improvement to be published at ICLR instead of a computational biology journal. Why not make the prior adaptive based on the data? There is opportunity to solve this problem that wouldn't require hand-crafting a new prior, or, make it more general for different datasets that would make it an _excellent_ contribution.

The paper's prose is clear. I do think that some of the tables could be rearranged to give the reader more clear insights into how the method works, which I relay below.

**Strengths:**

- Concrete biological problem of predicting FPTs that was well-motivated and explained.
- Clarity in explanation of method used.
- Clear advantage on benchmarks compared to other current solutions.
- Dramatic decrease in folding time.
- Demonstrated applicability to problems that also exhibit FPT characteristics.

**Weaknesses:**

- I would have liked to have seen Table 6 comparing all the different methods in the main paper. Plus, I would have liked to have seen comparison on MAE across methods, including in Table 1.
- Reliance on mulitmodal Gaussians for the prior.
- Reliance on only two metrics to measure accuracy of the metric.

**Questions:**

- Why not show the comparison between the sequence length and FPT in Table 1? I think that might provide more insight as to where KinPFN works better over other methods, unless sequence length isn't an important variable, which it seems to be since it's the subject of study in Figure 3.
- Why not put the legend in Figure 5 in the appendix? I'm not going to be able to copy/paste that to check, anyways.
- Why not use the KS test between CDFs as another comparison? This would help capture maximum discrepancy and add nuance to the experimental analysis.

---

> ### Author Response · Authors · 2024-11-21
> **Response to Reviewer PRsD**
>
> Dear Reviewer PRsD.
>
> Thank you for your valuable Feedback. We address your questions and concerns in the following.
>
> >I would have liked to have seen Table 6 comparing all the different methods in the main paper. Plus, I would have liked to have seen comparison on MAE across methods, including in Table 1
>
> We thank the reviewer for this useful comment and moved all results from Table 6 to the main paper. Additionally, we add MAE and Kolmogorov-Smirnov (KS) statistic results for the respective experiments in Appendix H.2 due to space limitations in the main body.
>
> >Reliance on multimodal Gaussians
>
> We agree with the reviewer that different distributions might further improve KinPFNs performance. We think that we adequately address this limitation in the discussion on future work in Section 6. However, as the first deep learning approach for RNA folding kinetics, our results indicate that assuming a Gaussian distribution for first passage times is reasonable and already leads to strong performance. Nevertheless, we plan to explore the potential of alternative distributional assumptions in future work.
>
> >Reliance on only two metrics to measure accuracy of the metric.
>
> The negative log-likelihood is commonly used to assess the performance of PFNs across different tasks and we think that it captures the performance of KinPFN arguably well. However, we are happy to include additional metrics that offer new insights or increase the understanding of strengths and weaknesses of our approach. For the specific case, we added MAE and KS to the analysis of the performance of KinPFN (results shown in Appendix H.2).
>
> >Why not show the comparison between the sequence length and FPT in Table 1? I think that might provide more insight as to where KinPFN works better over other methods unless sequence length isn't an important variable, which it seems to be since it's the subject of study in Figure 3
>
> We are currently analyzing the influence of sequence length along with other RNA features for all predictions and will add the requested analysis to the appendix, as we think that Table 1 might be overloaded otherwise. The number of potential structural states increases exponentially with the sequence length of the RNA. The reviewer, therefore, is right that simulations for longer RNAs require substantially longer runtimes of the simulators and are particularly challenging. As a pure in-context learner, KinPFN’s performance, however, is independent of the sequence length (similar to fitting KDEs or GMMs), which is one of the major advantages of our approach because it offers substantial speed-ups specifically for the case of long RNAs. The analysis shown in Figure 3 was performed to confirm this independence claim for KinPFN.
>
> >Why not put the legend in Figure 5 in the appendix? I'm not going to be able to copy/paste that to check, anyway.
>
> We included the three sequences in the figure legend to clarify that this experiment compares three different RNAs, each folding into the same secondary structure but undergoing distinctly different folding processes and efficiencies. We, therefore, disagree with the reviewer that the sequences provided in the legend of Figure 5 are invaluable for understanding the results and would like to keep the respective figure as it is.
>
> >Why not use the KS test between CDFs as another comparison? This would help capture maximum discrepancy and add nuance to the experimental analysis.
>
> We thank the reviewer for this helpful suggestion and we add the KS statistic results to Appendix H.2 in the revised version of our manuscript.
>
> We hope that we addressed all your questions and would like to thank the reviewer again for the valuable feedback. We are happy to answer further questions. If all your concerns have been addressed in the response, we would like to kindly ask you to increase our score.
>
> With kind regards,
>
> The authors.

---

### Official Review · Reviewer_TUhM · 2024-11-04

**Soundness:** 4
**Presentation:** 4
**Contribution:** 3
**Rating:** 6
**Confidence:** 4

**Summary:**

The paper introduces a model for predicting RNA folding dynamics called KinPFN. Specifically, this work focuses on computing the time needed for an RNA to fold into a specific structure (i.e., the first passage time). KinPFN uses a prior-data fitted network (PFN) that is calibrated using a sample of synthetic first passage time to predict entire cumulative distribution or RNA structures. The methodology is benchmarked on random and real sequences, and the paper concludes with an illustration of an application to gene expression.

Overall, the methodology is sound, and the results are convincing. The development of RNA folding dynamics prediction tools is timely and only a limited set of programs are currently available. Furthermore, machine learning approaches seem suited to this task, and, to my knowledge, this contribution is the first of its kind. The contributed approach is still a proof-of-concept, but it provides solid ground for further exploration of ML. The manuscript is clear and well written, yet it could benefit of clarifications suggested below.

**Strengths:**

Overall, it is a nice paper from a bioinformatics perspective. The machine learning component is a bit limited but I believe fits the broad definition of originality and significance through "application to a new domain." The authors basically apply one existing framework (prior-data filtering networks or PFNs) to model RNA folding dynamics. The innovative aspect of this work is thus rather the application to RNA folding. The research is timely, and the gain in speed could lead to interesting RNA design applications. The authors mention it but do not discuss it much. However, even if it is still very early, there are interesting/promising applications at the end of the paper. It is a clean and solid piece of work from which could emerge highly useful applications.

**Weaknesses:**

•	As far as I understand it, the structural model is based on secondary structures and uses (for simulations) the nearest neighbor (NN) energy model. The application to secondary structure is only partially described in the paper. It is an important limitation, and I am concerned that readers may miss this information. I would suggest updating Fig. 2 (reproduced from Muller et al, 2022) to include more details on the sampling (e.g., using a secondary structure model and kinfold) and eventually on the benchmark too. This is just a suggestion as the author(s) may prefer to include this information at other places of the manuscript (e.g., expand the background section to describe the folding model).
•	A claim of the paper is to drastically accelerate folding simulation. I agree this is a nice feature, but I also wonder to what extent it is currently a major bottleneck (for biologists) or what new applications it will enable. I think the manuscript could benefit from further justifications/motivations. For instance, the usefulness of KinPFN for design applications sounds very promising, and I would have liked more discussion (or experiments?) related to this topic. Moreover, since the models currently use kinfold simulations, it could be limited by the accuracy of the NN energy model, which has limitations on longer RNAs. Maybe KinPFN offers new perspectives to deal with this challenge?
•	It is not clear to me that PFNs perform much better than other options (e.g., KDEs) but at least they seem to be suited to the task. Table 1 compares KinPFN to DP-GMM and KDE models. The performance of KDEs seem very close to KingPFN, and I wonder if the author(s) could provide more context in the main body to discuss the significance of the results (e.g., from supp mat H.2)?
•	Fig. 3a shows a distribution of the performance across RNAs of various lengths. It may be out of the scope of this manuscript, but I would be interested to understand better how other features such as the nucleotide composition, MFE value, number of base pairs or loops, etc. impact the results. Also, how does KinPFN perform on multi-stable RNAs?

**Questions:**

Is there any further conceptual novelty in this submission that I might be missing?

---

> ### Author Response · Authors · 2024-11-21
> **Response to Reviewer TUhM**
>
> Dear Reviewer TUhM,
>
> Thank you for your valuable feedback and for highlighting the soundness and novelty of our approach. In the following, we provide answers to your questions and address your concerns in detail.
>
> >Is there any further conceptual novelty in this submission that I might be missing?
>
> We would like to further emphasize an additional unique aspect of our method: its ability to operate PFNs without requiring quantile information. Unlike traditional PFN approaches that predict the PPD of a target y for a given quantile x conditional on a dataset D, our method learns the entire PPD of y (in this case, the first passage times) without knowledge about the quantiles (x is always a zero-vector), effectively representing the absence of further information.
>
> This approach is motivated by the underlying problem structure, where we do not have access to the true quantiles of the context first-passage times. Therefore, we cannot treat the task as a standard regression problem but directly learn the PPD of first passage times conditional on a (data)set of context first passage times but without requiring quantile information which is novel in the field of PFNs.
>
> We add a small paragraph to the end of the background section to explicitly mark this change.
>
> >As far as I understand it, the structural model is based on secondary structures and uses (for simulations) the nearest neighbor (NN) energy model. The application to secondary structure is only partially described in the paper. It is an important limitation, and I am concerned that readers may miss this information. I would suggest updating Fig. 2 (reproduced from Muller et al, 2022) to include more details on the sampling (e.g., using a secondary structure model and Kinfold) and eventually on the benchmark too. This is just a suggestion as the author(s) may prefer to include this information at other places of the manuscript (e.g., expand the background section to describe the folding model).
>
> We agree with the reviewer that we do not discuss the secondary structure aspects in detail in the initial manuscript. However, we design KinPFN to be independent of the simulator, and therefore also of the underlying secondary structure folding model. While we do not want to speculate about it without empirical evidence, we think that KinPFN as a pure in-context learner would also be applicable to very different simulation data. In this regard, our experiments with Kfold simulations, different start and stop structures (both Section 5.1), as well as the application to gene expression data (Section 5.4) could be seen as the first evidence.
>
> Does this clarify the reviewers' concerns?
>
> >A claim of the paper is to dramatically accelerate folding simulation. I agree this is a nice feature, but I also wonder to what extent it is currently a major bottleneck (for biologists) or what new applications it will enable. I think the manuscript could benefit from further justifications/motivations For instance, the usefulness of KinPFN for design applications sounds very promising, and I would have liked more discussion (or experiments?) related to this topic.
>
> We agree with the reviewer that the speed-ups achieved by KinPFN are a major contribution of our work. These accelerations indeed offer the opportunity for novel applications, in particular but not limited to, the field of RNA design. However, while we also agree that further assessment of this strength would be very interesting, we also think that the development of a kinetic RNA design algorithm (and the required experiments) is out of the scope of this work. That said, a different application of KinPFN is shown in the case study on the folding efficiency of different RNAs in Section 5.3. In this regard, KinPFN could e.g. be used to identify switching states in RNAs. Nevertheless, we agree with the reviewer that we could discuss the potential benefits of our approach more in our manuscript and we add a short section to the Introduction of the revised version of the paper.

---

> ### Author Response · Authors · 2024-11-21
> **Response to Reviewer TUhM continued**
>
> >Moreover, since the models currently use Kinfold simulations, it could be limited by the accuracy of the NN energy model, which has limitations on longer RNAs. Maybe KinPFN offers new perspectives to deal with this challenge?
>
> We agree with the reviewer that KinPFN offers new perspectives to deal with limitations of the underlying simulators, particularly those limitations that are connected to sequence length which requires substantially longer simulations. Since KinPFN is a pure in-context learner with the simulation's first passage times as context, we cannot improve the accuracy of the NN energy model. However, we can significantly reduce the time required to get accurate approximations, completely independent of the sequence length. With our experiments across different sequence lengths (Section 5.1), we took a first approach in this direction, however, the experiment is limited to RNAs with a length of < 150nt due to the very long runtimes of the simulators.
>
> In addition, we would like to emphasize that we also use Kfold simulations (see Section 5.1) and achieve similar performance. This indicates that KinPFN’s predictions are independent of the underlying data-generating simulator but only depend on the provided context. We, therefore, expect KinPFN to work similarly well across different simulators which do not necessarily have to be secondary structure-based. This, however, is currently not empirically validated and requires further evaluations and larger-scale data acquisition.
>
> Does this answer the reviewer's question?
>
> >The performance of KDEs seem very close to KinPFN, and I wonder if the author(s) could provide more context in the main body to discuss the significance of the results (e.g., from supp mat H.2)
>
> We agree with the reviewer that, while outperformed by KinPFN, the performance of KDEs appears close to KinPFN. Nevertheless, we think that KinPFN has several advantages over KDEs, most obviously, it is not limited to multi-modal Gaussians for the formulation of the prior. As already stated by the reviewer, KinPFN is an early pioneering approach and there is still room for improvement. However, we will likely explore further applications of KinPFN for RNA kinetics in the future, including different distributions for the prior formulation. Regarding the explicit results of KDEs, we add a more comprehensive analysis including different metrics to Appendix H.2 and update the discussion of the results in the main body.
>
> >It may be out of the scope of this manuscript, but I would be interested to understand better how other features such as the nucleotide composition, MFE value, number of base pairs or loops, etc. impact the results
>
> We are preparing the requested analysis and will add it to the Appendix.
>
> However, we do not expect substantial changes in KinPFN’s ability to predict FPTs for sequences that have particularly low MFEs or exceptionally high GC contents. RNA structure is highly dependent on the sequence context but we do not expect any bias toward a particular shape of the FPT distributions as a result of the sequence/structure traits mentioned by the reviewer. While simple, our prior seems to cover a broad range of possible FPT distributions, and thus these traits should not have a strong influence on KinPFN’s prediction quality. Nevertheless, we are also curious to see the aforementioned results and will share them here when available.
>
> >How does KinPFN perform on multi-stable RNAs
>
> Multi-stable RNAs would typically produce CDF data with one or more plateaus, depending on the modalities of the FPT distribution. Since we train KinPFN with up to five modalities, the behavior of multi-stable RNAs is generally captured by our prior. Given a set of context first-passage times of a multi-stable RNA, we thus expect KinPFN to achieve similar performance as demonstrated in the paper.
>
> We hope that our responses clarified all the questions of the reviewer and we are happy to answer further questions if necessary. If all your concerns were addressed, we would be very thankful if you would increase your score.
>
> With kind regards,
>
> The authors

---

### Author Response · Authors · 2024-11-15
**Initial author response**

We thank all reviewers for their useful comments and valuable feedback. Specifically, we thank the reviewers for pointing out the novelty of our approach and its timeliness.

We will prepare individual responses for each review in the next few days.

We are looking forward to fruitful discussions and an interesting rebuttal period.

With kind regards,

The authors

---

### Author Response · Authors · 2024-11-21
**New revision of manuscript**

We update our manuscript to include the requested changes of the reviewers.

All changes in the text are highlighted in red.

We note that we are currently above the ten pages page limit due to the additional text and Figures.

We will ensure this limit for a potential CRC by moving some of the floats and text to the appendix if necessary.

Best regards,

The authors

---

### Author Response · Authors · 2024-11-26
**Revised Version of the manuscript**

We updated our manuscript to fit the page limit again. We further updated the draft of the overview Figure 1 with a full version of the figure. Due to the new figure, and additional text in response to the reviewers’ questions, we had to move the figure showing examples of the prior into the appendix.

If the reviewers have any other suggestions for structuring our manuscript, we are happy to hear them.

With kind regards,

The Authors

---

### Author Response · Authors · 2024-11-28
**Revised version of the manuscript**

We are happy that we finally obtained simulation data for two more RNA types, a SAM Riboswitch and a microRNA, as requested by reviewer DNAm. We updated our manuscript accordingly, adding approximations for these RNAs to the appendix.

With kind regards

The Authors

---

### Meta-Review · Area_Chair_Tis7 · 2024-12-21

**Metareview:**

The paper introduces a novel ML approach which employs prior-data fitted networks to compute RNA first passage times. The approach can be combined with RNA kinetics simulators to achieve significant speedup and has the potential to enable analyses that could not be performed previously due to prohibitive runtimes.

The presented method is the first ML-based approach for the problem of RNA first passage times. The effectiveness of the method is convincingly demonstrated. The authors have done a great job at addressing the reviewers' points, including clearly articulating the novel methodological aspects of the proposed approach, providing additional results (the first passage time distribution approximations on two additional RNA types), comparison of KINPFN against ensemble of GMMs, and other clarifying points.

Overall this is a great work that opens up exciting avenues for future ML research on the important task of approximating RNA first passage times.

**Additional Comments On Reviewer Discussion:**

The reviewers raised points regarding the need to clarify novel methodological aspects if applicable (concern share by Reviewers TUhM and DNAm), the focus on kinetics based on secondary structures (again a point raised by both reviewers TUhM and DNAm), providing additional discussion on the potential benefits of the approach, reporting additional results (KS test between CDFs, testing on additional RNA types requested by DNAm), comparison against GMM ensemble and other clarifying points requested by Reviewer FZT1.

The authors have addressed all these points in a very convincing way and have modified their manuscript accordingly. It is also remarkable that the authors were able to get hold of simulation data and provided approximation results for two additional RNA types during the rebuttal time.

---

### Decision · Program_Chairs · 2025-01-22

Accept (Poster)